



# Hydrological variations of the intermediate water masses of the western Mediterranean Sea during the past 20 ka inferred from neodymium isotopic composition in foraminifera and cold-water corals

Quentin Dubois-Dauphin[1], Paolo Montagna[2,3], Giuseppe Siani[1], Eric Douville[4], Claudia Wienberg[5], Dierk Hebbeln[5], Zhifei Liu[6], Nejib Kallel[7], Arnaud Dapoigny[4], Marie Revel[8], Edwige Pons-Branchu[4], Christophe Colin[1]*

[1]Laboratoire Geosciences Paris-Sud (GEOPS), Université de Paris Sud, Université Paris-Saclay, 91405 Orsay, France.
[2]ISMAR-CNR, via Gobetti 101, 40129 Bologna, Italy.
[3]Lamont-Doherty Earth Observatory, Columbia University, 61 Route 9W, Palisades, NY 10964, USA
[4]Laboratoire des Sciences du Climat et de l'Environnement, LSCE/IPSL, CEA-CNRS-UVSQ, Université Paris-Saclay, F-91191 Gif-sur-Yvette, France.
[5]MARUM-Center for Marine Environmental Sciences, University of Bremen, Leobener Strasse, 28359 Bremen, Germany.
[6]State Key Laboratory of Marine Geology, Tongji University, Shanghai 200092, China.
[7]Laboratoire Georessources, Matériaux, Environnements et Changements Globaux, LR13ES23, Faculté des Sciences de Sfax, Université de Sfax, BP1171, 3000 Sfax, Tunisia.
[8]Geoazur, UNS, IRD, OCA, CNRS, 250 rue Albert Einstein, 06500 Valbonne, France

*Correspondence to*: Christophe Colin (christophe.colin@u-psud.fr)

**Abstract.** The neodymium isotopic composition ($\varepsilon$Nd) of mixed planktonic foraminifera species and scleractinian cold-water corals (CWC; *Madrepora oculata, Lophelia pertusa*) collected at 280-620 m water depth in the Balearic Sea, the Alboran Sea and the south Sardinian continental margin was investigated to constrain hydrological variations at intermediate depths in the western Mediterranean Sea during the last 20 ka. Planktonic (*Globigerina bulloides*) and benthic (*Cibicidoides pachyderma*) foraminifera were also analyzed for stable oxygen ($\delta^{18}$O) and carbon ($\delta^{13}$C) isotopes. The foraminiferal and coral $\varepsilon$Nd values from the Balearic Sea and the Alboran Sea are comparable over the past ~13 ka, with mean values of -8.94±0.26 (1$\sigma$; n=24) and -8.91±0.18 (1$\sigma$; n=25), respectively. Before 13 ka BP, the foraminiferal $\varepsilon$Nd values are slightly lower (-9.28±0.15) and tend to reflect a higher mixing between intermediate and deep waters, characterized by more unradiogenic $\varepsilon$Nd values. The slight $\varepsilon$Nd increase after 13 ka BP is associated to a marked difference in the benthic foraminiferal $\delta^{13}$C composition of intermediate and deeper depths, which started at ~16 ka BP. This suggests an earlier stratification of the water masses and a subsequent reduced contribution of unradiogenic $\varepsilon$Nd from deep waters. The CWC from the Sardinia Channel show a much larger scattering of $\varepsilon$Nd values, from -8.66±0.30 to -5.99±0.50, and a lower average (-7.31±0.73; n=19) compared to the CWC and foraminifera from the Alboran Sea and Balearic Sea, indicative of intermediate waters sourced from the Levantine basin. At the time of sapropel S1 deposition (10.2 to 6.4 ka), the $\varepsilon$Nd values of the Sardinian CWC become more unradiogenic (-8.38±0.47; n=3 at ~8.7 ka BP), suggesting a significant contribution of intermediate waters originated from the western basin. Accordingly, we propose here that western Mediterranean intermediate waters replaced the Levantine Intermediate Water (LIW), which was strongly reduced during the mid-sapropel (~8.7 ka BP). This



observation supports a notable change of Mediterranean circulation pattern centered on sapropel S1 that needs
further investigations to be confirmed.

**1. Introduction**
The Mediterranean Sea is a mid-latitude semi-enclosed basin, characterized by evaporation exceeding
precipitation and river runoff, where the inflow of fresh and relatively warm surface Atlantic water is
transformed into saltier and cooler (i.e. denser) intermediate and deep waters. Several studies have demonstrated
that the Mediterranean thermohaline circulation was highly sensitive to both the rapid climatic changes
propagated into the basin from high latitudes of the Northern Hemisphere (Cacho et al., 1999, 2000, 2002;
Moreno et al., 2002, 2005; Paterne et al., 1999; Martrat et al., 2004; Sierro et al., 2005; Frigola et al., 2007,
2008) and orbitally-forced modifications of the eastern Mediterranean freshwater budget mainly driven by
monsoonal river runoff from the south (Rohling et al., 2002; 2004; Bahr et al., 2015). A link between the
intensification of the Mediterranean Outflow Water (MOW) and the intensity of the Atlantic Meridional
Overturning Circulation (AMOC) was proposed (Cacho et al., 1999, 2000, 2001; Bigg and Wadley, 2001; Sierro
et al., 2005; Voelker et al., 2006) and recently supported by new geochemical data in sediments of the Gulf of
Cádiz (Bahr et al., 2015). In particular, it has been suggested that the intensity of the MOW and, more generally,
the variations of the thermohaline circulation of the Mediterranean Sea could play a significant role in triggering
a switch from a weakened to an enhanced state of the AMOC through the injection of saline Mediterranean
waters in the intermediate North Atlantic at times of weak AMOC (Rogerson et al., 2006; Voelker et al., 2006;
Khélifi et al., 2009). Since the Mediterranean intermediate waters, notably the Levantine Intermediate Water
(LIW), represent today up to 80 % in volume of the MOW (Kinder and Parilla, 1987) and are therefore a key
driver of MOW-derived salt into the North Atlantic, it is crucial to gain a more complete understanding of the
variability of the Mediterranean intermediate circulation in the past and its impact on the outflow.
Previous studies have mainly focused on the glacial variability of the deep-water circulation in the western
Mediterranean basin (Cacho et al., 2000, 2006; Sierro et al., 2005; Frigola et al., 2007, 2008). During the Last
Glacial Maximum (LGM), strong deep-water convection took place in the Gulf of Lions, producing cold, well-
ventilated western Mediterranean Deep Water (WMDW) (Cacho et al., 2000, 2006; Sierro et al., 2005), while
the MOW flowed at greater depth in the Gulf of Cádiz (Rogerson et al., 2005; Schönfeld and Zahn, 2000). With
the onset of the Termination 1 (T1) at about 15 ka, the WMDW production declined until the transition to the
Holocene due to the rising sea level, with a relatively weak mode during the Heinrich Stadial 1 (HS1) and the
Younger Dryas (YD) (Sierro et al., 2005; Frigola et al., 2008), that led to the deposition of the Organic Rich
Layer 1 (ORL1; 14.5-8.2 ka BP; Cacho et al., 2002).
Because of the disappearance during the Early Holocene of specific epibenthic foraminiferal species, such as
*Cibicidoides* spp., which are commonly used for paleohydrological reconstructions, information about the
Holocene variability of the deep-water circulation in the western Mediterranean are relatively scarce and are
mainly based on grain size analysis and sediment geochemistry (Frigola et al., 2007). These authors have
identified four distinct phases representing different deep-water overturning conditions in the western
Mediterranean basin during the Holocene, as well as centennial- to millennial-scale abrupt events of overturning
reinforcement.



Faunal and stable isotope records from benthic foraminifera located at intermediate depths in the eastern basin
reveal uninterrupted well-ventilated LIW during the last glacial period and deglaciation (Kuhnt et al., 2008;
Schmiedl et al., 2010). A grain-size record obtained from a sediment core collected within the LIW depth range
(~500 m water depth) at the east Corsica margin also reveals enhanced bottom currents during HS1 and the YD
(Toucanne et al., 2012). The Early Holocene is characterized by a collapse of the LIW (Kuhnt et al., 2008;
Schmiedl et al., 2010; Toucanne et al., 2012) synchronous with the sapropel S1 deposition (10.2 – 6.4 cal ka BP;
Mercone et al., 2000). Proxies for deep-water conditions reveal the occurrence of episodes of deep-water
overturning reinforcement in the eastern Mediterranean basin at 8.2 ka BP (Rohling et al., 1997, 2015; Kuhnt et
al., 2007; Abu-Zied et al., 2008, Siani et al., 2013; Tachikawa et al; 2015), responsible for the interruption of the
sapropel S1 in the eastern Mediterranean basin (Mercone et al., 2001; Rohling et al., 2015).
It has recently been shown that the neodymium (Nd) isotopic composition, expressed as εNd =
$([(^{143}Nd/^{144}Nd)_{sample} /(^{143}Nd/^{144}Nd)_{CHUR}] - 1)$ x 10000 (CHUR: Chondritic Uniform Reservoir [Jacobsen and
Wasserburg , 1980]) of living and fossil scleractinian CWC faithfully traces intermediate and deep-water mass
provenance and mixing of the ocean (e.g. van de Flierdt et al., 2010; Colin et al., 2010; López Correa et al.,
2012; Monterro-Serrano et al., 2011, 2013; Copard et al., 2012). Differently from the CWC, the εNd
composition of fossil planktonic foraminifera is not related to the ambient seawater at calcification depths but
reflects the bottom and/or pore water εNd, due to the presence of authigenic Fe-Mn coatings precipitated on their
carbonate shell (Roberts et al., 2010; Elmore et al., 2011; Piotrowski et al., 2012; Tachikawa et al., 2014; Wu et
al., 2015). Therefore, the εNd composition of planktonic foraminiferal tests can be used as a useful tracer of
deep-water circulation changes in the past, although the effect of pore water on foraminiferal εNd values could
potentially complicate the interpretation (Tachikawa et al., 2014).
In the Mediterranean Sea, modern seawater εNd values display a large range from ~-11 to ~-5, and a clear
vertical and longitudinal gradient, with more radiogenic values encountered in the eastern basin and typically at
intermediate and deeper depths (Spivack and Wasserburg 1988; Henry et al., 1994; Tachikawa et al., 2004;
Vance et al., 2004). Considering this large εNd contrast, εNd recorded in fossil CWC and planktonic
foraminifera from the Mediterranean offers great potential to trace intermediate and deep-water mass exchange
between the two basins, especially during periods devoid of key epibenthic foraminifera, such as the sapropel S1
and ORL1 events.
Here, the εNd of planktonic foraminifera from a sediment core collected in the Balearic Sea and CWC samples
from the Alboran Sea and the Sardinia Channel was investigated to establish past changes of the εNd values at
intermediate depths and constrain hydrological variations of the LIW during the past ~20 ka. The εNd values
have been combined with stable oxygen ($\delta^{18}$O) and carbon ($\delta^{13}$C) isotope measurements of benthic (*Cibicidoides*
*pachyderma*) and planktonic (*Globigerina bulloides*) foraminifera and sea-surface temperature estimates by
modern analogue technique (MAT). Results reveal significant changes of the E-W gradient of εNd values for the
LIW of the western basin interpreted by a drastic reduction of the hydrological exchanges between the western
and eastern Mediterranean Sea and the subsequent higher proportion of intermediate water produced in the Gulf
of Lions during the time interval corresponding to the sapropel S1 deposition.





### 2. Seawater εNd distribution in the Mediterranean Sea

The Atlantic Water (AW) enters the Mediterranean Sea as surface inflow through the Strait of Gibraltar with an
unradiogenic εNd signature of ~-9.7 in the strait (Tachikawa et al., 2004) and ~-10.4 in the Alboran Sea
(Tachikawa et al., 2004, Spivack and Wasserburg, 1988) for depths shallower than 50 m. During its eastward
flowing, AW mixes with upwelled Mediterranean Intermediate Water forming the Modified Atlantic Water
(MAW) that spreads within the basin (Millot and Taupier-Letage, 2005) (Fig.1). The surface water εNd values
(shallower than 50 m) range from -9.8 to -8.8 in the western Mediterranean basin (Henry et al., 1994; Montagna
et al., in prep) and -9.3 to -4.2 in the eastern basin, with seawater off the Nile delta showing the most radiogenic
values (Tackikawa et al., 2004; Vance et al., 2004; Montagna et al., in prep). The surface waters in the eastern
Mediterranean basin become denser due to strong mixing and evaporation caused by cold and dry air masses
flowing over the Cyprus-Rhodes area in winter, and eventually sink leading to the formation of LIW
(Ovchinnikov, 1984; Lascaratos et al., 1993, 1998; Malanotte-Rizzoli et al., 1999; Pinardi and Masetti, 2000).
The LIW spreads throughout the entire Mediterranean basin at depths between ~150-200 m and ~600-700 m,
and is characterized by more radiogenic εNd values ranging from -7.9 to -4.8 (average value ± 1σ: -6.6 ± 1) in
the eastern basin and from -10.4 to -7.58 (-8.7 ± 0.9) in the western basin (Henry et al., 1994; Tachikawa et al.,
2004; Vance et al., 2004; Montagna et al., in prep). The LIW acquires its εNd signature mainly from the partial
dissolution of Nile River particles (Tachikawa et al., 2004), which have an average isotopic composition of -3.25
(Weldeab et al., 2002), and the mixing along its path with overlying and underlying water masses with different
εNd signatures. The LIW finally enters the Atlantic Ocean at intermediate depths through the Strait of Gibraltar
with an average εNd value of -9.2 ± 0.2 (Tachikawa et al., 2004; Montagna et al., in prep).
The WMDW is formed in the Gulf of Lions due to winter cooling and evaporation followed by mixing between
the relative fresh surface water and the saline LIW and spreads into the Balearic basin and Tyrrhenian Sea
between ~2000 m and 3000 m (Millot, 1999; Schroeder et al., 2013) (Fig. 1). The WMDW is characterized by an
average εNd value of -9.4 ± 0.9 (Henry et al., 1994; Tachikawa et al., 2004; Montagna et al., in prep). Between
the WDMW and the LIW (from ~700 to 2000 m), the Tyrrhenian Deep Water (TDW) has been found (Millot et
al., 2006), which is produced by the mixing between WMDW and Eastern Mediterranean Deep Water (EMDW)
that cascades in the Tyrrhenian Sea after entering from the Strait of Sicily (Millot, 1999, 2009; Astraldi et al.,
2001). The TDW has an average εNd value of -8.1 ± 0.5 (Montagna et al., in prep).

### 3. Material and methods

*3.1. Cold-water coral and foraminifera samples*
Forty-four CWC samples belonging to the species *Lophelia pertusa* and *Madrepora oculata* collected from the
Alboran Sea and the Sardinia Channel were selected for this study (Fig. 1). Nineteen fragments were collected at
various core depths from a coral-bearing sediment core (RECORD 23; 38°42.18' N; 08°54.75' E; Fig. 1)
retrieved from 414 m water depth in the "Sardinian Cold-Water Coral Province" (Taviani et al., 2015) during the
R/V Urania cruise "RECORD" in 2013. The Sardinian CWC samples were used for U-series dating and Nd
isotopic composition measurements. For the southern Alboran Sea, twenty-five CWC samples were collected at
water depths between 280 and 442 m in the "eastern Melilla Coral Province" (Fig. 1) during the R/V Poseidon
cruise "POS-385" in 2009 (Hebbeln et al. 2009). Eleven samples were collected at the surface of two coral
mounds (New Mound and Horse Mound) and three coral ridges (Brittlestar ridges I, II and III), using a box corer



and a remotely operated vehicle (ROV). In addition, fourteen CWC samples were collected from various core
depths of three coral-bearing sediment cores (GeoB13728, 13729 and 13730) retrieved from the Brittlestar ridge
I. Details on the location of surface samples and cores collected in the southern Alboran Sea and details on the
radiocarbon ages obtained from these coral samples are reported in Fink et al. (2013). Like the CWC sample set
from the Sardinia Channel, the dated Alboran CWC samples were also used for further Nd isotopic composition
analyses in this study.
In addition, a sediment core (barren of any CWC fragments) was recovered in the Balearic Sea at 622 m water
depth during the R/V Le Suroît cruise "PALEOCINAT II" in 1992 (SU92-33; 35°25.38' N; 0°33.86' E; Fig. 1).
The core was sub-sampled continuously at 5-10 cm intervals for the upper 2.1 m for a total number of 24
samples used for further multi-proxy analyzes.

**3.2. Analytical procedures on cold-water coral samples**
*3.2.1. U/Th dating*
The nineteen CWC samples collected from the sediment core RECORD 23 (Sardinia Channel) were analysed for
uranium and thorium isotopes to obtain absolute dating using a Thermo Scientific$^{TM}$ Neptune$^{Plus}$ MC-ICPMS
installed at the Laboratoire des Sciences du Climat et de l'Environnement (LSCE, Gif-sur-Yvette, France). Prior
to analysis, the samples were carefully cleaned using a small diamond blade to remove any visible contamination
and sediment-filled cavities. The fragments were examined under a binocular microscope to ensure against the
presence of bioeroded zones and finally crushed into a coarse-grained powder with an agate mortar and pestle.
The powders (~60-100 mg) were transferred to acid cleaned Teflon beakers, ultrasonicated in MilliQ water,
leached with 0.1N HCl for ~ 15 s and finally rinsed twice with MilliQ water. The physically and chemically
cleaned samples were dissolved in 3-4 ml dilute HCl (~10%) and mixed with an internal triple spike with known
concentrations of $^{229}$Th, $^{233}$U and $^{236}$U, calibrated against a Harwell Uraninite solution (HU-1) assumed to be at
secular equilibrium. The solutions were evaporated to dryness at 70°C, redissolved in 0.6 ml 3N HNO$_3$ and then
loaded into 500 µl columns packed with Eichrom UTEVA resin to isolate uranium and thorium from the other
major and trace elements of the carbonate matrix. The U and Th separation and purification followed a
procedure slightly modified from Douville et al. (2010). The U and Th isotopes were determined following the
protocol recently revisited at LSCE (Pons-Branchu et al., 2014). The $^{230}$Th/U ages were calculated from
measured atomic ratios through iterative age estimation (Ludwig and Titterington, 1994), using the $^{230}$Th, $^{234}$U
and $^{238}$U decay constants of Cheng et al. (2013) and Jaffey et al. (1971). Due to the low $^{232}$Th concentration (< 1
ng/g; see Table 1), no correction was applied for the non-radiogenic $^{230}$Th fraction.

*3.2.2 Nd isotopic composition analyses on cold-water coral fragments*
Sub-samples of the CWC fragments from the Sardinia Channel used for U-series dating in this study (Table 1) as
well as sub-samples of the twenty-five CWC fragments originating from the Alboran Sea, which were already
radiocarbon-dated by Fink et al. (2013) (Table 2), were used for further Nd isotopic composition analyses. The
fragments (350 to 600 mg) were subjected to a mechanical and chemical cleaning procedure. The visible
contaminations, such as Fe-Mn coatings and detrital particles, were carefully removed from the inner and
outermost surfaces of the coral skeletons using a small diamond blade. The physically cleaned fragments were
ultrasonicated for 10 min with 0.1 N ultra-clean HCl, followed by several MilliQ water rinses and finally





dissolved in 2.5 N ultraclean $HNO_3$. Nd was separated from the carbonate matrix using Eichrom TRU and LN
resins, following the analytical procedure described in detail in Copard et al. (2010).
The $^{143}Nd/^{144}Nd$ ratios of all purified Nd fractions were analyzed using the ThermoScientific Neptune[Plus] Multi-
Collector Inductively Coupled Plasma Mass Spectrometer (MC-ICP-MS) hosted at LSCE. The mass-
fractionation correction was made by normalizing $^{146}Nd/^{144}Nd$ to 0.7219 and applying an exponential law.
During each analytical session, samples were systematically bracketed with analyses of JNdi-1 and La Jolla
standard solutions, which are characterised by accepted values of 0.512115±0.000006 (Tanaka et al., 2000) and
0.511855±0.000007 (Lugmair et al., 1983), respectively. Standard JNdi-1 and La Jolla solutions were analysed
at concentrations similar to those of the samples (5-10 ppb) and all the measurements affected by instrumental
bias were corrected, when necessary, using La Jolla standard. The external reproducibility (2σ) for time resolved
measurement, deduced from repeated analyses of La Jolla and JNdi-1 standards, ranged from 0.1 to 0.5 εNd
units for the different analytical sessions. The analytical error for each sample analysis was taken as the external
reproducibility of the La Jolla standard for each session. Concentrations of Nd blanks were negligible compared
to the amount of Nd of CWC investigated in this study.

**3.3. Analyses on sediment of core SU92-33**
*3.3.1. Radiocarbon dating*
Radiocarbon dating was measured at UMS-ARTEMIS (Pelletron 3MV) AMS (CNRS-CEA Saclay, France).
Seven AMS radiocarbon ($^{14}C$) dating were performed in core SU92-33 on well-preserved calcareous tests of the
planktonic foraminifera *G. bulloides* in the size fraction >150 μm (Table 3). The age model for the core was
derived from the calibrated planktonic ages by applying a mean reservoir effect of ~400 years (Siani et al., 2000,
2001). All $^{14}C$ ages were converted to calendar years (cal. yr BP, BP = AD 1950) by using the INTCAL13
calibration data set (Reimer et al., 2013) and the CALIB 7.0 program (Stuiver and Reimer, 1993).

*3.3.2. Stable isotopes*
Stable oxygen ($\delta^{18}O$) and carbon ($\delta^{13}C$) isotope measurements were performed in core SU92-33 on well-
preserved (clean and intact) samples of the planktonic foraminifera *G. bulloides* (250-315 μm fraction) and the
epibenthic foraminifera *C. pachyderma* (250-315 μm fraction) using a Finnigan MAT-253 mass spectrometer at
the State Key Laboratory of Marine Geology (Tongji University). Both $\delta^{18}O$ and $\delta^{13}C$ values are presented
relative to the Pee Dee Belemnite (PDB) scale by comparison with the National Bureau of Standards (NBS) 18
and 19. The mean external reproducibility was checked by replicate analyses of laboratory standards and is better
than ±0.07‰ (1σ) for $\delta^{18}O$ and ±0.04‰ for $\delta^{13}C$.
*3.3.3 Nd isotope measurements on planktonic foraminifera*
Approximately 25 mg of mixed planktonic foraminifera species were picked from the >63 μm size fraction of
each sample already used for stable isotope measurements (Table 4). The samples were gently crushed between
glass slides under the microscope to ensure that all chambers were open, and ultrasonicated with MilliQ water.
Samples were allowed to settle between ultrasonication steps before removing the supernatant. Each sample was
rinsed thoroughly with MilliQ water until the solution was clear and free of clay. The cleaned samples were
dissolved in 1N acetic acid and finally centrifuged to ensure that all residual particles were removed, following



the procedure described in Roberts et al. (2010). Nd was separated following the analytical procedure reported in
Wu et al. (2015). For details on the measurement of Nd isotopes see the section above.

247        *3.3.4. Modern analogue technique (MAT)*

The palaeo-sea surface temperatures (SST) were estimated using the modern analogue technique (MAT)
(Hutson, 1980; Prell, 1985), implemented by Kallel et al. (1997) for the Mediterranean Sea. This method directly
measures the difference between the faunal composition of a fossil sample with a modern database, and it
identifies the best modern analogues for each fossil assemblage (Prell, 1985). Reliability of SST reconstructions
is estimated using a square chord distance test (dissimilarity coefficient), which represents the mean degree of
similarity between the sample and the best 10 modern analogues. When the dissimilarity coefficient is lower than
0.25, the reconstruction is considered to be of good quality (Overpeck et al., 1985; Kallel et al., 1997). For core
SU92-33, good dissimilarity coefficients are <0.2, with an average value of 0.13.

257        **4.    Results**

258        ***4.1.    Cold-water coral ages***

The good state of preservation for the CWC samples from the Sardinia Channel (RECORD 23; Fig. 1) is attested
by their initial $\delta^{234}$U values (Table 1), which is in the range of the modern seawater value (146.8±0.1; Andersen
et al., 2010). If the uncertainty of the $\delta^{234}$U$_i$ is taken into account, all the values fulfill the so-called "strict" ± 4
‰ reliability criterion and the U/Th ages can be considered strictly reliable. The coral ages range from
0.091±0.011 to 10.904±0.042 ka BP (Table 1), and reveal three distinct clusters of coral age distribution during
the Holocene representing periods of sustained coral occurrence. These periods coincide with the Early Holocene
encompassing a 700-years-lasting time interval from ~10.9 to 10.2 ka BP, the very late Early Holocene at ~8.7
ka BP, and the Late Holocene starting at ~1.5 ka BP (Table 1).
Radiocarbon ages obtained for CWC samples collected in the Alboran Sea were published by Fink et al. (2013)
(Table 2). They also document three periods of sustained CWC occurrence coinciding with the Bølling–Allerød
(B-A) interstadial (13.5–12.9 cal ka BP), the Early Holocene (11.2–9.8 cal ka BP) and the Mid- to Late Holocene
(5.4–0.3 cal ka BP).

272        ***4.2 Chronological framework for core SU92-33***

273        The stratigraphy of core SU92-33 was derived from the $\delta^{18}$O variations of the planktonic foraminifera

*G. bulloides* (Fig. 2b). The last glacial/interglacial transition and the Holocene encompasses the upper 2.1 m of
the core (Fig. 2b). The $\delta^{18}$O record of *G. bulloides* shows higher values (~3.5 ‰) during the late glacial
compared to the Holocene (from ~1.5 to 0.8 ‰) exhibiting a pattern similar to those observed in nearby deep-sea
cores from the Western Mediterranean Sea (Sierro et al., 2005; Melki et al., 2009).

278        The age model of core SU92-33 is based on 7 AMS-$^{14}$C age measurements for the upper 1.2 m of the core

and by a linear interpolation between these ages (Table 3, Fig. 2). Below, a control point has been established for
the onset of the last deglaciation that presents a coeval age in the western and central Mediterranean Sea at about
17 cal ka BP (Sierro et al., 2005; Melki et al., 2009; Siani et al., 2001). The upper 2.1 m of core SU92-33 spans
the last 19 ka, with an estimated average sedimentation rate between 9 to 15 cm ka$^{-1}$, with the lowest values
observed during the Holocene.



### 4.3 SST reconstructions of core SU92-33

April-May SST reconstruction was derived from MAT to define the main climatic events recorded in core SU92-33 during the last 19 ka. SSTs vary from 8.5°C to 17.5°C with high amplitude variability over the last 19 ka BP (Fig. 2a). The LGM (19-18 ka BP) is characterized by SST values centered at around 12°C. Then, a progressive decrease of ~4°C between 17.8 ka to 16 ka marks the Henrich Stadial 1 (HS1) (Fig. 2a). A warming phase (~14°C) between 14.5 ka BP and 13.8 ka BP coincides with the B-A interstadial and is followed by a cooling (~11°C) between 13.1 ka BP and 11.8 ka BP largely corresponding to the YD (Fig. 2). During the Holocene, SSTs show mainly values of ~16°C, with one exception between 7 ka BP and 6 ka BP pointing to an abrupt cooling of ~3°C (Fig. 2a). From the late glacial to the Holocene, SST variations show a similar pattern to that previously observed in the Gulf of Lions and Tyrrhenian Sea (Kallel et al., 1997; Melki et al., 2009) and globally synchronous for the main climatic transitions to the well dated South Adriatic Sea core MD90-917 confirming the robustness of the SU92-33 age model (Fig. 2a).

### 4.4 Benthic stable oxygen and carbon isotope records of core SU92-33

The $\delta^{18}O$ and $\delta^{13}C$ records obtained from the benthic foraminifera *C. pachyderma* display significant variations at millennial time scales (Figs. 2c and 2d). The $\delta^{18}O$ values decrease steadily from ~4.5 ‰ during the LGM to ~1.5 ‰ during the Holocene, without showing any significant excursion during HS1 and the YD events (Fig. 2c), in agreement with results obtained for the neighbor core MD99-2343 (Sierro et al., 2005).

The $\delta^{13}C$ record obtained from *C. pachyderma* shows a decreasing trend since the LGM with a low variability from ~1.6 ‰ to ~0.6 ‰ (Fig. 2d). The heaviest $\delta^{13}C$ values are related to the LGM (~1.6 ‰) while the lightest values (~0.6 ‰) characterize the Early Holocene and in particular the period corresponding to the sapropel S1 event in the eastern Mediterranean basin (Fig. 2d).

### 4.5 Nd isotopic composition of planktonic foraminifera and cold-water corals

εNd values of planktonic foraminifera of core SU92-33 collected from the Balearic Sea vary within a relatively narrow range between -9.50±0.30 and -8.61±0.30, with an average value of -9.06±0.28 (Table 2, Fig. 3b). The record shows a slight increasing trend since the LGM, with the more unradiogenic values (average -9.28±0.15; n=7) observed in the oldest part of the record (between 18 and 13.5 ka BP), whereas Holocene values are generally more radiogenic (average -8.84±0.22; n=17) (Fig. 3b).

The εNd record obtained for the CWC samples from the Alboran Sea displays a narrow range from -9.22±0.30 to -8.59±0.3, which is comparable to the εNd record obtained on planktonic foraminifera from the Balearic Sea over the last 13.5 ka (Table 2, Fig. 3b). Most of the CWC εNd values are similar within error and the record does not reveal any clear difference over the last ~13.5 ka.

Finally, the CWC samples from the Sardinia Channel display εNd values ranging from -5.99±0.50 to -7.75±0.10 during the Early and Late Holocene, and values as low as -8.66±0.30 during the the mid-sapropel S1 deposition (S1a) (~8.7 ka BP) (Table 1, Fig. 3c).



## 5. Discussion

As first observations, the CWC and foraminiferal εNd values measured for this study indicate a pronounced dispersion at intermediate depth in terms of absolute values and variability in Nd isotopes during the Holocene between the Alboran and Balearic Seas and the Sardinia Channel. In addition the foraminiferal εNd record reveals an evolution towards more radiogenic values at intermediate water depth in the Balearic Sea over the last ~19 ka (Fig. 3).

A prerequisite to properly interpret such εNd values differences and variations through time consists in characterizing the present-day εNd of the main water-mass end-members circulating in the western Mediterranean basin. This is possible by evaluating the temporal changes in εNd of the end-members since the LGM, and assessing the potential influences of lithogenic Nd input and regional exchange between the continental margins and seawater ("boundary exchange"; Lacan and Jeandel, 2001, 2005) on the εNd values of intermediate water masses.

During its westward flow, the LIW continuously mixes with surrounding waters with different εNd signatures lying above and below. For the western Mediterranean basin, these water masses are the MAW/Western Intermediate Water (WIW) and the TDW/WMDW, respectively. Accordingly, a well-defined and gradual εNd gradient exists at intermediate depth between the eastern and western Mediterranean basins, with LIW values becoming progressively more unradiogenic towards the Strait of Gibraltar, from -4.8±0.2 at 227 m in the Levantine basin to -10.4±0.2 at 200 m in the Alboran Sea (Tachikawa et al., 2004). Such an εNd paatern implies an effective vertical mixing with more unradiogenetic water masses along the E-W LIW trajectory ruling out severe isotopic modifications of the LIW due to the local exchange between the continental margins and seawater. Unfortunately, no information exists on the potential temporal variability in εNd of the Mediterranean water-mass end-members since the LGM.

It has been demonstrated that eolian dust input can modify the surface and sub-surface εNd distribution of the ocean in some areas (Arsouze et al., 2009). The last glacial period was associated with an aridification of North Africa (Sarnthein et al., 1981; Hooghiemstra et al., 1987; Moreno et al., 2002; Wienberg et al., 2010) and higher fluxes of Saharan dust to the NE tropical Atlantic (Itambi et al., 2009) and the western Mediterranean Sea characterized by unradiogenic εNd values (between -10±0.4 and -17±0.4; Grousset et al., 1992, 1998; Grousset and Biscaye, 2005). Bout-Roumazeilles et al. (2013) documented a dominant role of eolian supply in the Siculo-Tunisian Strait during the last 20 ka, with the exception of a significant riverine contribution (from the Nile River) and a strong reduction of eolian input during the sapropel S1 event. Such variations in the eolian input to the Mediterranean Sea are not associated to a significant change in the seawater εNd record obtained for the Balearic Sea (core SU92-33) during the sapropel S1 event (Fig. 3). Furthermore, the εNd signature of the CWC from the Sardinia Channel (core RECORD 23) shifts to more unradiogenic values (-8.66±0.30) during the sapropel S1 event, which is opposite to what expected if related to a strong reduction of eolian sediment input. Thus, these results suggest that changes of eolian dust input since the LGM were not responsible for the observed εNd variability at intermediate water depths.

Consequently, assuming that the Nd isotopic budget of the western Mediterranean Sea has not been strongly modified since the LGM, the reconstructed variations of the E-W gradient of εNd values in the western Mediterranean Sea for the past and notably during the sapropel S1 event (Fig. 3) are indicative of a major reorganization of intermediate water circulation.



### 5.1 Hydrological changes in the Alboran and Balearic Seas since the LGM

The range in εNd for the CWC from the Alboran Sea (from -9.22±0.30 to -8.8.59±0.30; Table 2) is very close to the one obtained for the planktonic foraminifera from the Balearic Sea (from -9.50±0.3 to -8.61±0.3; Table 4, Fig. 3c), suggesting that both sites are influenced by the same intermediate water masses at least for the last 13.5 ka BP. Today, LIW occupies a depth range between ~200 and ~700 m in the western Mediterranean basin (Millot, 1999; Sparnocchia et al., 1999). More specifically, the salinity maximum corresponding to the core of LIW is found at around 400 m in the Alboran Sea (Millot, 2009) and up to 550 m in the Balearic Sea (López-Jurado et al., 2008). The youngest CWC sample collected in the Alboran Sea with a rather "recent" age of 0.34 cal ka BP (Fink et al. 2013) displays an εNd value of -8.59±0.30 (Table 2) that is similar to the present-day value of the LIW at the same site (-8.3±0.2) (Dubois-Dauphin et al., submitted) and is significantly different from the WMDW εNd signature in the Alboran Sea (-10.7±0.2, 1270 m water depth; Tachikawa et al., 2004). Considering the intermediate depth range of the studied CWC and foraminifera samples, we can reasonably assume that samples from both sites, in the Balearic Sea (622 m water depth) and in the Alboran Sea (280 to 442 m water depth), record εNd variations of the LIW. The εNd record obtained on planktonic foraminifera generally displays more unradiogenic and homogenous values before ~13 cal ka BP (range: -9.46 to -9.12) compared to the most recent part of the record (range: -9.50 to -8.61), with the highest value of -8.61±0.3 in the Early and Late Holocene.

The $\delta^{18}O$ record obtained on *G. bulloides* indicates an abrupt 1‰ excursion towards lighter values centered at about 16 cal ka BP (Table 4), synchronous with the HS1 (Fig. 2b), which is similar to the $\delta^{18}O$ shift reported by Sierro et al. (2005) for a core collected at 2391 m water depth NE of the Balearic Islands (MD99-2343; Fig. 1). As the Heinrich events over the last glacial period are characterized by colder and fresher surface water in the Alboran Sea (Cacho et al., 1999; Pérez-Folgado et al., 2003; Martrat et al., 2004) and dry climate on land over the western Mediterranean Sea (Allen et al., 1999; Combourieu-Nebout et al., 2002; Sanchez Goni et al., 2002; Bartov et al., 2003), lighter $\delta^{18}O$ values of planktonic *G. bulloides* are thought to be the result of the inflow of freshwater derived from the melting of icebergs in the Atlantic Ocean into the Mediterranean Sea (Sierro et al., 2005; Rogerson et al., 2008).

During this time interval, the $\delta^{13}C$ record of *C. pachyderma* from the Balearic Sea (core SU92-33) displays a decreasing $\delta^{13}C$ trend after ~16 cal ka BP (from 1.4 ‰ to 0.9 ‰; Table 4; Fig. 4a). Moreover, the $\delta^{13}C$ record obtained on benthic foraminifera *C. pachyderma* from the deep Balearic Sea (core MD99-2343) reveals similar $\delta^{13}C$ values before ~16 cal ka BP suggesting well-mixed and ventilated water masses during the LGM and the onset of the deglaciation (Sierro et al., 2005).

The slightly lower foraminiferal εNd values before ~13 cal ka BP could reflect a stronger influence of water masses deriving from the Gulf of Lions as WMDW (εNd: -9.4±0.9; Henry et al., 1994; Tachikawa et al., 2004; Montagna et al., in prep). This is in agreement with εNd results obtained by Jiménez-Espejo et al. (2015) from planktonic foraminifera collected from deep-water sites (1989 m and 2382 m) in the Alboran Sea (Fig. 4c). Jiménez-Espejo et al. (2015) documented lower εNd values (ranging from -10.14±0.27 to -9.58±0.22) during the LGM, suggesting an intense deep-water formation. This is also associated to an enhanced activity of the deeper branch of the MOW in the Gulf of Cádiz (Rogerson et al., 2005; Voelker et al., 2006) linked to the active production of the WMDW in the Gulf of Lions during the LGM (Jiménez-Espejo et al., 2015).





The end of the HS1 (14.7 cal ka BP) is concurrent with the onset of the B-A warm interval characterized by
increased SST identified for various sites in the Mediterranean Sea (Cacho et al., 1999; Martrat et al., 2004;
Essallami et al., 2007), in agreement with the SST record obtained for the Balearic Sea (SU92-33: Fig. 3a). The
B-A interval is associated to the so-called melt-water pulse 1A (e.g. Weaver et al., 2003) occurring at around
14.5 cal ka BP. This led to a rapid sea-level rise of about 20 m in less than 500 years and large freshwater
discharges in the Atlantic Ocean due to the melting of continental ice sheets (Deschamps et al., 2012), resulting
in an enhanced Atlantic inflow across the Strait of Gibraltar. Synchronously, cosmogenic dating of Alpine
glacier retreat throughout the western Mediterranean hinterland suggests maximum retreat rates (Ivy-Ochs et al.,
2007; Kelly et al., 2006). Overall, these events are responsible for freshening Mediterranean waters and reduced
surface water density, and hence, weakened ventilation of intermediate (Toucanne et al., 2012) and deep-water
masses (Cacho et al., 2000; Sierro et al., 2005). Similarly, lower benthic $\delta^{13}C$ values obtained for the Balearic
Sea (Fig. 4a) point to less ventilated intermediate water relative to the late glacial. In addition, a decoupling in
the benthic $\delta^{13}C$ values is observed between deep (MD99-2343) and intermediate (core SU92-33) waters after
~16 cal ka BP (Sierro et al. 2005), suggesting an enhanced stratification of the waters masses (Fig. 4a). At this
time, the shallowest εNd record from the deep Alboran Sea (core 300G) shifted towards more radiogenic values,
while the deepest one (core 304G) remained close to the LGM values (Jimenez-Espejo et al., 2015) (Fig. 4c).
Furthermore, results from the UP10 fraction (particles > 10 μm) of the MD99-2343 sediment core (Fig. 4d),
indicate a declining bottom-current velocity at 15 ka (Frigola et al., 2008). Rogerson et al. (2008) have
hypothesized that during deglacial periods the sinking depth of dense waters produced in the Gulf of Lions was
shallower resulting in new intermediate water (WIW) rather than new deep-water (WMDW) as observed today
during mild winters (Millot, 1999; Schott et al., 1996). Therefore, intermediate depths of the Balearic Sea could
have been isolated from the deep-water with the onset of the T1 (at ~15 ka BP). The reduced convection in the
deep western Mediterranean Sea together with the shoaling of the nutricline (Rogerson et al., 2008) led to the
deposition of the ORL 1 (14.5 to 8.2 ka B.P; Cacho et al., 2002) and dysoxic conditions below 2000 m in
agreement with the absence of epibenthic foraminifera such as *C. pachyderma* after 11 cal ka BP in MD99-2343
(Sierro et al., 2005) (Fig. 4a).
After 13.5 ka BP, planktonic foraminifera εNd values from the Balearic Sea (core SU92-33) become more
radiogenic and are in the range of CWC εNd values from the Alboran Sea (Fig. 4b). These values may reveal a
stronger influence of the LIW in the Balearic Sea during the Younger Dryas, as also supported by the sortable
silt record from the Tyrrhenian Sea (Toucanne et al., 2012) (Fig. 4e). Deeper depths of the Alboran Sea also
record a stronger influence of the LIW with an εNd value of -9.1±0.4 (Jimenez-Espejo et al., 2015). In addition,
a concomitant activation of the upper MOW branch, as reconstructed from higher values of Zr/Al ratio in
sediments of the Gulf of Cádiz, can be related to the enhanced LIW flow in the western Mediterranean Sea (Fig.
4f) (Bahr et al., 2015).
The time of sapropel S1 deposition (10.2 – 6.4 ka) is characterized by a weakening or a shutdown of
intermediate- and deep-water formation in the eastern Mediterranean basin (Rossignol-Strick et al., 1982; Cramp
and O'Sullivan, 1999; Emeis et al., 2000; Rohling et al., 2015). At this time, planktonic foraminifera εNd values
from intermediate water depths in the Balearic Sea (core SU92-33) remain high (between -9.15±0.3 and -
8.61±0.3) (Fig. 4b). On the other hand, the deeper Alboran Sea provides a value of -9.8±0.3 pointing to a





stronger contribution of WMDW (Jimenez-Espejo et al., 2015), coeval with the recovery of deep-water activity
from core MD99-2343 (Frigola et al., 2008).

*5.2 Hydrological changes in the Sardinia Channel during the Holocene*
The present-day hydrographic structure of the Sardinia Channel is characterized by four water masses, with the
surface, intermediate and deep-water masses being represented by MAW, LIW and TDW/WMDW, respectively
(Astraldi et al., 2002a; Millot and Taupier-Lepage, 2005). In addition, the WIW, flowing between the MAW and
the LIW, has also been observed along the Channel (Sammari et al., 1999). The core of the LIW is located at
400-450 m water depth in the Tyrrhenian Sea (Hopkins, 1988; Astraldi et al., 2002b), which is the depth range of
CWC samples from the Sardinia Channel (RECORD 23; 414 m) (Taviani et al., 2015). The youngest CWC
sample dated at ~0.1 ka BP has an εNd value of -7.70±0.10 (Table 1, Fig. 5), which is similar within error to the
value obtained from a seawater sample collected at 451 m close to the coral sampling location (-8.0±0.4;
Montagna et al., in prep).
The CWC dating from the Sardinia Channel shows three distinct periods of sustained coral occurrence in this
area during the Holocene, with each displaying a large variability in εNd values. CWC from the Early Holocene
(10.9-10.2 ka BP) and the Late Holocene (<1.5 ka BP) exhibit similar ranges of εNd values (ranging from -
5.99±0.50 to -7.75±0.20; Table 1, Fig 5c). Such variations are within the present-day εNd range being
characteristic for intermediate waters in the eastern Mediterranean Sea (-6.6±1.0; Tachikawa et al., 2004; Vance
et al., 2004). However, the CWC εNd values are more radiogenic than those observed at mid-depth in the
present-day western basin (ranging from -10.4±0.2 to -7.58 ±0.47; Henry et al., 1994; Tachikawa et al., 2004;
Montagna et al., in prep), suggesting a stronger LIW component in the Sardinia Channel during the Early and
Late Holocene. The Sardinian CWC εNd variability also reflects the sensitivity of the LIW to changes in the
eastern basin such as rapid variability of the Nile River flood discharge (Revel et al., 2014; 2015; Weldeab et al.,
2014) or a modification through time in the proportion between the LIW and the Cretan Intermediate Water
(CIW). Today, the intermediate water outflowing from the Strait of Sicily is composed by ~66 to 75 % of LIW
and 33 to 25 % of CIW (Manca et al., 2006; Millot, 2014). As the CIW is formed in the Aegean Sea, this
intermediate water mass is generally more radiogenic than LIW (Tachikawa et al., 2004; Montagna et al., in
prep). Following this hypothesis, a modification of the mixing proportion between the CIW and the LIW may
potentially explain values as radiogenic as about -6 in the Sardinia Channel during the Early and Late Holocene
(Fig. 5c). However, a stronger LIW and/or a CIW contribution cannot be responsible for εNd values as low as -
8.66±0.30 observed during the sapropel S1 event at 8.7 ka BP (Table 1, Fig. 5c). Considering that such
unradiogenic value is not observed at intermediate depth in the modern eastern Mediterranean basin, the most
plausible hypothesis suggested here is that the CWC were bathed in intermediate waters which were more
marked by the western basin.

*5.3 Hydrological implications for the intermediate water masses of the western Mediterranean Sea*
The εNd records of the Balearic Sea, Alboran Sea and Sardinia Channel document a temporal variability of the
east-west gradient in the western Mediterranean basin during the Holocene. The magnitude of the gradient
ranges from ~1.5 to ~3 ε units during the Early and Late Holocene and it is strongly reduced at 8.7 ka BP,





coinciding with the sapropel S1 event affecting the eastern Mediterranean basin (Fig. 5). Such variations could be the result of a modification of the Nd isotopic composition of intermediate water masses due to intensity changes of the the LIW through time and a higher contribution of the western-sourced intermediate water towards the Sardinia Channel coinciding with the sapropel S1 event.

The LIW acquires its radiogenic εNd in the Mediterranean Levantine basin mainly from Nd exchange between seawater and lithogenic particles originating mainly from Nile River (Tachikawa et al., 2004). A higher sediment supply from the Nile River starting at ~15 ka BP was documented by a shift to more radiogenic εNd values of the terrigenous fraction obtained from a sediment core having been influenced by the Nile River discharge (Revel et al., 2015) (Fig. 5e). However, others studies pointed to a gradual enhanced Nile River runoff as soon as 14.8 ka and a peak of Nile discharge from 9.7 to 8.4 ka recorded by large increase in sedimentation rate from 9.7 to 8.4 ka (>120 cm/ka) (Revel et al., 2015; Weldeab et al., 2014; Castaneda et al., 2016). The increase in Nile River discharge has been related to the African Humid Period (14.8–5.5 ka BP; Shanahan et al., 2015), which in turn was linked to the precessional increase in Northern Hemisphere insolation during low eccentricity (deMenocal et al., 2000; Barker et al., 2004; Garcin et al., 2009). An increasing amount of radiogenic sediments dominated by the Blue/Atbara Nile River contribution (Revel et al., 2014) could have modified the εNd of surface water towards more radiogenic values (Revel et al., in prep). This signature was likely transferred to intermediate depth as a consequence of the LIW formation in the Rhodes Gyre, and it might have been propagated westwards towards the Sardinia Channel.

The Nile River runoff was also strongly enhanced during the sapropel S1 event (Revel et al., 2010; Weldeab et al., 2014; Revel et al., 2014). Scrivner et al. (2004) have reported very high foraminifera εNd values (-3 to -3.5) corresponding to the sapropel S1 event in the eastern Levantine Basin (ODP site 967; 34°04.27′N, 32°43.53′E; 2553 m water depth), pointing to a maximum Nile discharge at this time. Hence, considering the more unradiogenic value of the CWC samples from the Sardinia Channel during the sapropel S1a event, it is very unlikely that eastern-sourced water flowed at intermediate depth towards the Sardinia Channel. A possible explanation could be the replacement of the radiogenic LIW that was no longer produced in the eastern basin (Rohling, 1994) by less radiogenic western intermediate water (possibly WIW). Such a scenario could even support previous hypotheses that invoke a potential circulation reversal in the eastern Mediterranean from anti-estuarine to estuarine during sapropel formation (Huang and Stanley, 1972; Calvert, 1983; Sarmiento et al., 1988; Buckley and Johnson, 1988; Thunell and Williams, 1989).

**6. Conclusions**

The foraminiferal εNd record from the intermediate Balearic Sea reveals a relatively narrow range of εNd values varying between -9.50 and -8.61 since the LGM (~20 ka). Between 18 and 13.5 cal ka BP, the more unradiogenic εNd values support a vigorous deep overturning in the Gulf of Lions while $\delta^{18}O$ and $\delta^{13}C$ values indicate a stratification of the water masses after 16 cal ka BP. The stratification together with a decrease of the deep-water intensity led to more radiogenic values after ~13 cal ka BP. The εNd record from planktonic foraminifera, supplemented by CWC from the intermediate depths of the Alboran Sea, show only minor changes in εNd values from 13.5 cal ka BP to 0.34 cal ka BP, suggesting that the westernmost part of the western Mediterranean basin is not very sensitive to hydrological variations of the LIW.





On the contrary, CWC located at the depth of the LIW in the Sardinia Channel indicate high amplitude variations
of the εNd values (between -7.75±0.10 and -5.99±0.50) during the Holocene, which could highlight either the
role of the Nile River in changing the εNd of the LIW in the eastern Mediterranean basin or a different
LIW/CIW mixing of the water outflowing from the Strait of Sicily. Coinciding with the sapropel S1 event at
~8.7 ka BP, CWC display a shift toward lower values (-8.66±0.30), similar to those obtained at intermediate
depths in the westernmost part of the western basin. This suggests that western-sourced intermediate water likely
filled mid-depth of the southern Sardinia, replacing LIW that was no longer produced (or heavily reduced) in the
eastern basin. These results could potentially support a reversal of the Mediterranean circulation, although this
assumption needs further investigation to be confirmed.

**Acknowledgements**
The research leading to this study has received funding from the French National Research Agency
"Investissement d'Avenir" (n°ANR-10-LABX-0018), the HAMOC project ANR-13-BS06-0003, the
MISTRALS/PALEOMEX/COFIMED and ENVIMED/Boron Isotope and Trace Elements project. This work
contributes to the RITMARE project. We thank Hiske Fink for selecting and kindly providing the cold-water
corals samples from the Alboran Sea. We further thank François Thil and Louise Bordier for their support with
Nd isotopic composition analyses. Paolo Montagna is grateful for financial support from the Short Term
Mobility Program (CNR). Thanks are also extended to the captains, crews, chief scientists, and scientific parties
of research cruises RECORD (R/V Urania), POS-385 (R/V Poseidon) and PALEOCINAT II (R/V Le Suroît).

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

**Table captions**

**Table 1**. U-series ages and εNd values obtained for cold-water coral samples collected from sediment core RECORD 23
(Sardinia Channel).

**Table 2**. εNd values obtained for cold-water corals from the southern Alboran Sea. The AMS [14]C ages published by Fink et
al. (2013) are also reported as Median probability age (ka BP).

**Table 3**. AMS [14]C ages of samples of the planktonic foraminifer *G. bulloides* from 'off-mound' sediment core SU92-33. The
AMS [14]C ages were corrected for [13]C and a mean reservoir age of 400 yrs, and were converted into calendar years using the
INTCAL13 calibration data set (Reimer et al., 2013) and the CALIB 7.0 program (Struiver et al., 2005).

**Table 4**. Multiproxy data obtained for the upper 2.1 m of sediment core SU92-33 (Balearic Sea). Stable oxygen and carbon
isotopes were measured on benthic (*C. pachyderma*) and planktonic (*G. bulloides*) foraminifera; εNd values were obtained on
mixed planktonic foraminifera samples. The age results from a combination of 7 AMS-[14]C age measurements for the upper
1.2 m of the core and by a linear interpolation between these ages as well as the $\delta^{18}O$ variations of the planktonic
foraminifera *G. bulloides*.

**Figure captions**

**Figure 1**. Map of the western Mediterranean Sea showing the locations of samples investigated in this study. Yellow dot
indicates the sampling location of the sediment core from the Balearic Sea (SU92-33); yellow stars indicate the locations of
the CWC-bearing cores from the Sardinia Channel (RECORD 23) and the southern Alboran Sea (for further details on the
CWC from the Alboran Sea refer also to Fink et al., 2013). The cores discussed in this paper (Gulf of Cádiz: IODP site
U1387, Balearic Sea: MD09-2343, northern Tyrrhenian Sea: MD01-2472, Adriatic Sea: MD90-917) are indicated by black
dots, and seawater stations are marked by open squares. Arrows represent the main oceanographic currents. The black line
shows the general trajectory of the Modified Atlantic Water (MAW) flowing at the surface from the Atlantic Ocean toward
the western and eastern Mediterranean. The orange line represents the Levantine Intermediate Water (LIW) originating from
the eastern basin. The black dashed line shows the trajectory of the Western Mediterranean Deep Water (WMDW) flowing
from the Gulf of Lions toward the Strait of Gibraltar.

**Figure 2**. (a) Sea Surface Temperature (SST) records of cores SU92-33 (red line) and MD90-917 (green line), (b) $\delta^{18}O$
record obtained on planktonic foraminifer *G. bulloides* for core SU92-33, (c) $\delta^{18}O$ record obtained on benthic foraminifer *C.*
*pachyderma* for core SU92-33, (d) $\delta^{13}C$ record obtained on benthic foraminifer *C. pachyderma* for core SU92-33. LGM: Last





Glacial Maximum; HS1: Heinrich Stadial 1; B-A: Bølling-Allerød; YD: Younger Dryas. Black triangles indicate AMS $^{14}$C
age control points.

**Figure 3**. (a) Sea Surface Temperature (SST) record of core SU92-33 (red line), (b) εNd records obtained on mixed
planktonic foraminifers from core SU92-33 (open circles) and from cold-water coral fragments collected in the Alboran Sea
(red squares), (c) εNd values of cold-water corals from core RECORD 23 (Sardinia Channel).

**Figure 4**. (a) δ$^{13}$C records obtained on benthic foraminifer *C. pachyderma* for cores SU92-33 (red line) and MD99-2343
(blue line; Sierro et al., 2005). (b) εNd records obtained on mixed planktonic foraminifers from core SU92-33 (open circles)
and from cold-water coral fragments collected in the Alboran Sea (red squares). Modern εNd values for LIW (orange dashed
line) and WMDW (blue dashed line) are also reported for comparison. (c) εNd values obtained for planktonic foraminifera
with Fe-Mn coatings at sites 300G (36°21.532' N, 1°47.507' W; 1860 m; open dots) and 304G (36°19.873' N, 1°31.631' W;
2382 m; black dots) in Alboran Sea (Jimenez-Espejo et al., 2015). (d) UP10 fraction (>10 μm) from core MD99-2343
(Frigola et al., 2008). (e) Sortable silt mean grain-size of core MD01-2472 (Toucanne et al., 2012). (f) Ln Zr/Al ratio at IODP
site U1387 (36°48.3' N 7°43.1' W; 559 m) (Bahr et al., 2015).

**Figure 5.** (a) δ$^{18}$O record obtained on planktonic foraminifer *G. bulloides* for core SU92-33, (b) δ$^{13}$C records obtained on
benthic foraminifer *C. pachyderma* for core SU92-33, (c) εNd values of cold-water corals from core RECORD 23 (Sardinia
Channel), (d) εNd values records obtained on mixed planktonic foraminifera from core SU92-33 (open circles) and from
cold-water coral fragments collected in the Alboran Sea (red squares), (e) εNd values obtained on terrigenous fraction of
MS27PT located close the Nile River mouth in the eastern Mediterranean basin (Revel et al., 2015).

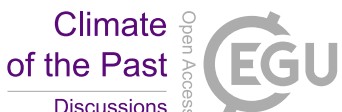



Figure 1





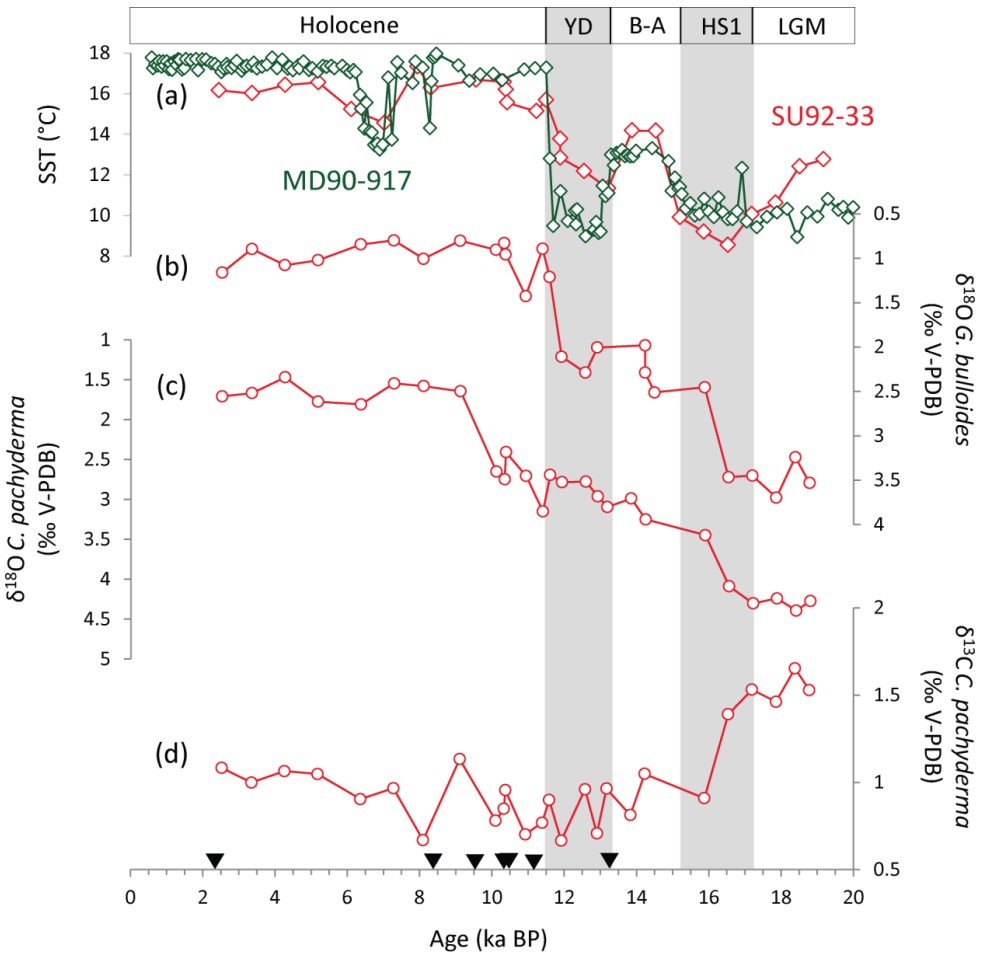

**Figure 2**





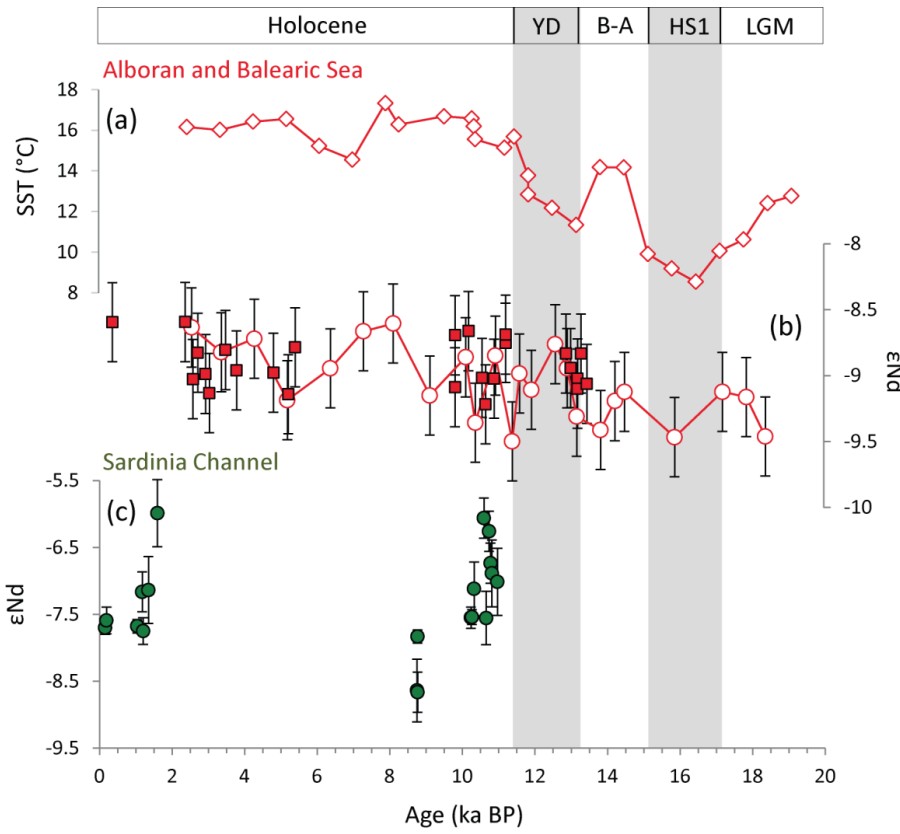

**Figure 3**





**Figure 4**



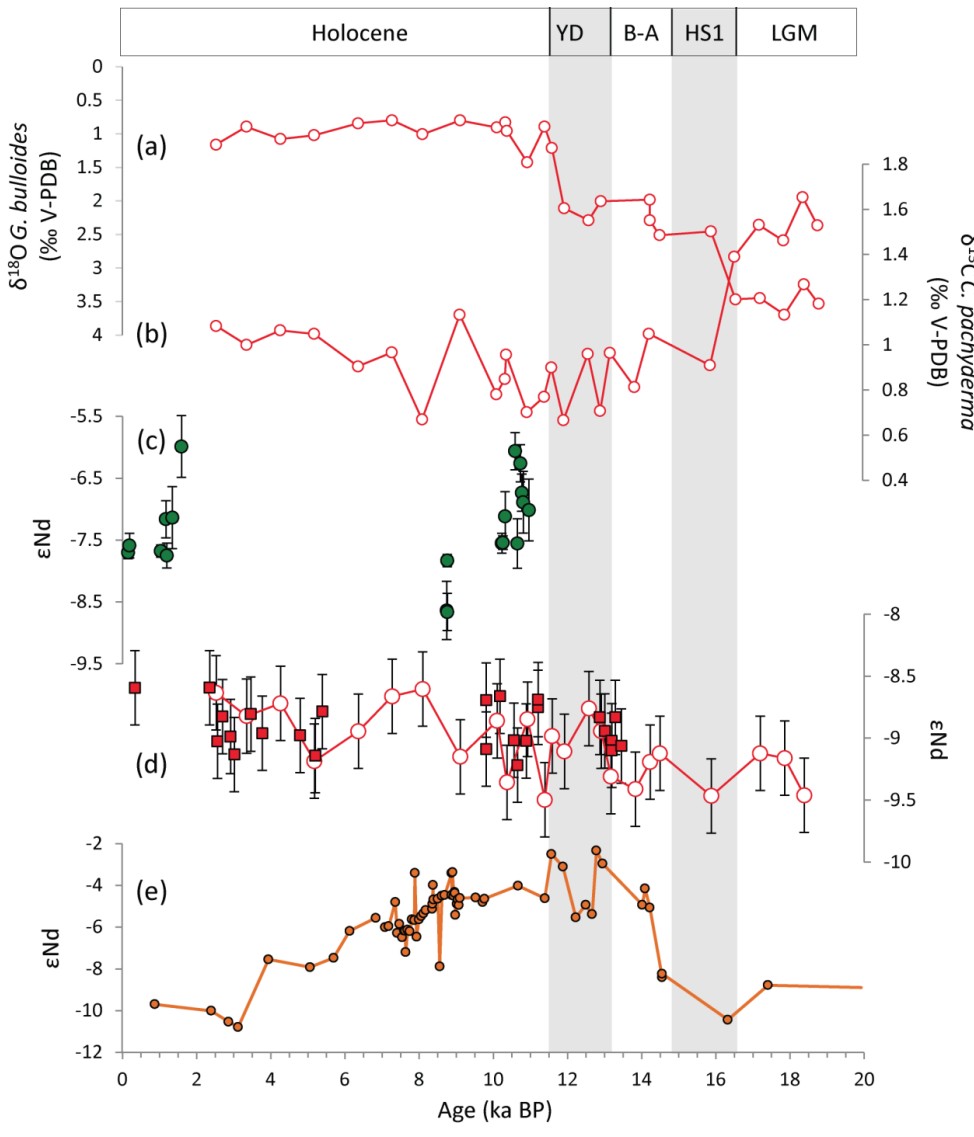

**Figure 5**



**Table 1**

| Sample ID | Depth in core (cm) | Corals species | $^{238}U$ (µg/g) | $^{232}Th$ (ng/g) | $\delta^{234}U_m$ (‰) | $^{230}Th/^{238}U$ | $^{230}Th/^{232}Th$ | Age (ka BP) | $\delta^{234}U_{(0)}$ (‰) | $^{143}Nd/^{144}Nd$ | εNd |
|---|---|---|---|---|---|---|---|---|---|---|---|
| RECORD_23_V | 0-3.5 | Madrepora oculata | 3.31 ±0.005 | 0.68 ±0.014 | 151.85 ±1.7 | 0.00163 ±0.00011 | 25 ±1.7 | 0.091 ±0.011 | 151.92 ±1.7 | 0.512243 ±0.000005 | -7.70 ±0.10 |
| RECORD_23_V | 3-7 | Madrepora oculata | 3.23 ±0.002 | 0.52 ±0.001 | 147.11 ±0.6 | 0.00199 ±0.00006 | 38 ±1.1 | 0.127 ±0.006 | 147.19 ±0.6 | 0.512249 ±0.000010 | -7.59 ±0.20 |
| RECORD_23_V | 7-10 | Madrepora oculata | 3.99 ±0.007 | 0.25 ±0.002 | 147.52 ±1.7 | 0.01227 ±0.00022 | 640 ±11.6 | 1.110 ±0.023 | 148.01 ±1.7 | 0.512244 ±0.000015 | -7.68 ±0.30 |
| RECORD_23_V | 8-10 | Madrepora oculata | 3.79 ±0.005 | 0.41 ±0.001 | 147.77 ±0.7 | 0.01253 ±0.00007 | 350 ±2.0 | 1.135 ±0.008 | 148.27 ±0.7 | 0.512271 ±0.000010 | -7.16 ±0.20 |
| RECORD_23_IV | 6-9 | Madrepora oculata | 4.06 ±0.006 | 0.35 ±0.001 | 148.47 ±1.2 | 0.01366 ±0.00011 | 480 ±3.8 | 1.243 ±0.012 | 149.02 ±1.2 | 0.512241 ±0.000010 | -7.75 ±0.20 |
| RECORD_23_IV | 27-30 | Madrepora oculata | 4.06 ±0.003 | 1.09 ±0.001 | 146.91 ±1.3 | 0.01405 ±0.00013 | 159 ±1.4 | 1.283 ±0.014 | 147.47 ±1.3 | 0.512272 ±0.000026 | -7.14 ±0.50 |
| RECORD_23_IV | 37-40 | Madrepora oculata | 3.52 ±0.005 | 0.08 ±0.000 | 148.25 ±1.1 | 0.01663 ±0.00012 | 2308 ±16.4 | 1.529 ±0.013 | 148.92 ±1.1 | 0.512331 ±0.000026 | -5.99 ±0.50 |
| RECORD_23_III | 55-57 | Madrepora oculata | 3.63 ±0.002 | 0.27 ±0.000 | 145.30 ±0.7 | 0.08832 ±0.00020 | 3530 ±8.1 | 8.685 ±0.027 | 148.93 ±0.8 | 0.512195 ±0.000026 | -8.64 ±0.50 |
| RECORD_23_III | 58-61 | Madrepora oculata | 4.24 ±0.004 | 0.36 ±0.001 | 146.71 ±1.2 | 0.08859 ±0.00037 | 3336 ±14.0 | 8.702 ±0.048 | 150.39 ±1.2 | 0.512237 ±0.000010 | -7.83 ±0.20 |
| RECORD_23_III | 63-66 | Lophelia pertusa | 4.15 ±0.005 | 0.42 ±0.002 | 147.19 ±0.8 | 0.08863 ±0.00054 | 2783 ±17.1 | 8.703 ±0.063 | 150.89 ±0.9 | 0.512194 ±0.000015 | -8.66 ±0.30 |
| RECORD_23_I | 0-2 | Lophelia pertusa | 3.35 ±0.002 | 0.37 ±0.000 | 147.02 ±0.7 | 0.10283 ±0.00018 | 2788 ±4.8 | 10.173 ±0.025 | 151.34 ±0.7 | 0.512251 ±0.000010 | -7.55 ±0.20 |
| RECORD_23_I | 62-65 | Lophelia pertusa | 3.27 ±0.003 | 0.39 ±0.002 | 144.75 ±1.2 | 0.10289 ±0.00061 | 2721 ±16.1 | 10.201 ±0.075 | 149.01 ±1.2 | 0.512251 ±0.000010 | -7.54 ±0.20 |
| RECORD_23_II | 50-52 | Lophelia pertusa | 2.92 ±0.003 | 0.92 ±0.003 | 145.39 ±1.6 | 0.10351 ±0.00061 | 1046 ±6.2 | 10.260 ±0.079 | 149.69 ±1.6 | 0.512273 ±0.000021 | -7.12 ±0.40 |
| RECORD_23_I | 12-14 | Lophelia pertusa | 3.07 ±0.002 | 0.49 ±0.000 | 145.22 ±0.7 | 0.10609 ±0.00023 | 1971 ±4.3 | 10.531 ±0.031 | 149.64 ±0.7 | 0.512327 ±0.000015 | -6.06 ±0.30 |
| RECORD_23_I | 5-7 | Lophelia pertusa | 3.50 ±0.002 | 0.42 ±0.000 | 146.35 ±0.9 | 0.10677 ±0.00016 | 2654 ±4.0 | 10.591 ±0.025 | 150.82 ±0.9 | 0.512251 ±0.000021 | -7.55 ±0.40 |
| RECORD_23_II | 94-98 | Lophelia pertusa | 3.14 ±0.003 | 0.62 ±0.002 | 146.42 ±1.0 | 0.10755 ±0.00047 | 1737 ±7.6 | 10.672 ±0.059 | 150.94 ±1.0 | 0.512317 ±0.000015 | -6.26 ±0.30 |
| RECORD_23_I | 15-17 | Lophelia pertusa | 3.40 ±0.003 | 0.46 ±0.000 | 146.01 ±0.9 | 0.10790 ±0.00021 | 2409 ±4.6 | 10.713 ±0.031 | 150.53 ±0.9 | 0.512293 ±0.000015 | -6.73 ±0.30 |
| RECORD_23_II | 96-100 | Lophelia pertusa | 3.61 ±0.004 | 0.35 ±0.001 | 145.50 ±0.8 | 0.10821 ±0.00044 | 3579 ±14.7 | 10.750 ±0.055 | 150.02 ±0.8 | 0.512285 ±0.000026 | -6.89 ±0.50 |
| RECORD_23_II | 93-95 | Lophelia pertusa | 3.19 ±0.003 | 0.24 ±0.000 | 143.33 ±0.8 | 0.10947 ±0.00032 | 4381 ±12.7 | 10.904 ±0.042 | 147.85 ±0.9 | 0.512279 ±0.000026 | -7.01 ±0.50 |





| Sample ID | Core depth (cm) | Species | Water Depth (m) | Median probability age (ka BP) | $^{143}$Nd/$^{144}$Nd | εNd |
|---|---|---|---|---|---|---|
| GeoB 13727-1#1 | Surface | *Lophelia pertusa* | 363 | 0.339 | 0.512198 ±0.000015 | -8.59 ±0.30 |
| GeoB 13727-1#2 | Surface | *Madrepora oculata* | 353 | 2.351 | 0.512198 ±0.000015 | -8.59 ±0.30 |
| GeoB 13730-1 | 6 | *Lophelia pertusa* | 338 | 2.563 | 0.512175 ±0.000015 | -9.03 ±0.30 |
| GeoB 13728-1 | Bulk (0-15) | *Lophelia pertusa* | 343 | 2.698 | 0.512185 ±0.000015 | -8.83 ±0.30 |
| GeoB 13728-2 | 2 | *Lophelia pertusa* | 343 | 2.913 | 0.512177 ±0.000015 | -8.99 ±0.30 |
| GeoB 13722-3 | Bulk (0-15) | *Madrepora oculata* | 280 | 3.018 | 0.512170 ±0.000015 | -9.13 ±0.30 |
| GeoB 13722-3 | Bulk (15-30) | *Madrepora oculata* | 280 | 3.463 | 0.512186 ±0.000015 | -8.81 ±0.30 |
| GeoB 13735-1 | Bulk (0-15) | *Madrepora oculata* | 280 | 3.770 | 0.512179 ±0.000015 | -8.96 ±0.30 |
| GeoB 13723-1 | Bulk (0-8) | *Madrepora oculata* | 291 | 4.790 | 0.512178 ±0.000015 | -8.98 ±0.30 |
| GeoB 13725-2 | Surface | *Madrepora oculata* | 355 | 5.201 | 0.512169 ±0.000015 | -9.14 ±0.30 |
| GeoB 13723-1 | Bulk (8-20) | *Madrepora oculata* | 291 | 5.390 | 0.512187 ±0.000015 | -8.79 ±0.30 |
| GeoB 13729-1 | 2.5 | *Lophelia pertusa* | 442 | 9.810 | 0.512172 ±0.000015 | -9.09 ±0.30 |
| GeoB 13729-1 | 2.5 | *Lophelia pertusa* | 442 | 9.810 | 0.512193 ±0.000015 | -8.69 ±0.30 |
| GeoB 13729-1 | 49 | *Lophelia pertusa* | 442 | 10.181 | 0.512194 ±0.000015 | -8.66 ±0.30 |
| GeoB 13730-1 | 102 | *Lophelia pertusa* | 338 | 10.556 | 0.512176 ±0.000015 | -9.02 ±0.30 |
| GeoB 13730-1 | 194 | *Lophelia pertusa* | 338 | 10.652 | 0.512165 ±0.000015 | -9.22 ±0.30 |
| GeoB 13729-1 | 315 | *Lophelia pertusa* | 442 | 10.889 | 0.512176 ±0.000015 | -9.02 ±0.30 |
| GeoB 13729-1 | 375 | *Lophelia pertusa* | 442 | 11.206 | 0.512189 ±0.000015 | -8.75 ±0.30 |
| GeoB 13730-1 | 298 | *Lophelia pertusa* | 338 | 11.208 | 0.512193 ±0.000015 | -8.69 ±0.30 |
| GeoB 13728-2 | 191 | *Lophelia pertusa* | 343 | 12.874 | 0.512185 ±0.000015 | -8.83 ±0.30 |
| GeoB 13737-1#2 | Surface | *Lophelia pertusa* | 297 | 13.005 | 0.512180 ±0.000015 | -8.94 ±0.30 |
| GeoB 13728-2 | 295 | *Lophelia pertusa* | 364 | 13.194 | 0.512176 ±0.000015 | -9.02 ±0.30 |
| GeoB 13728-2 | 295 | *Lophelia pertusa* | 364 | 13.194 | 0.512171 ±0.000015 | -9.10 ±0.30 |
| GeoB 13730-1 | 427 | *Lophelia pertusa* | 338 | 13.291 | 0.512185 ±0.000015 | -8.83 ±0.30 |
| GeoB 13737-1#1 | Surface | *Lophelia pertusa* | 299 | 13.452 | 0.512174 ±0.000015 | -9.06 ±0.30 |

**Table 2**





| Core | Depth in core (cm) | $^{14}$C-age (years) | ±1σ (years) | Median probability age (ka BP) |
|---|---|---|---|---|
| SU92-33 | 0 | 2770 | 70 | 2437 |
| SU92-33 | 64 | 7870 | 90 | 8280 |
| SU92-33 | 70 | 8670 | 80 | 9528 |
| SU92-33 | 74 | 9510 | 100 | 10295 |
| SU92-33 | 84 | 9610 | 90 | 10389 |
| SU92-33 | 90 | 10180 | 100 | 11192 |
| SU92-33 | 120 | 11710 | 110 | 13172 |

**Table 3**





| Depth in core (cm) | Age (ka BP) | δ¹³C C. pachyderma (‰ VPDB) | δ¹⁸O C. pachyderma (‰ VPDB) | δ¹³C G. bulloides (‰ VPDB) | δ¹⁸O G. bulloides (‰ VPDB) | ¹⁴³Nd/¹⁴⁴Nd | εNd | |
|---|---|---|---|---|---|---|---|---|
| 1 | 2.53 | 1.08 | 1.71 | -0.6 | 1.16 | 0.512195 ±0.000015 | -8.64 | ±0.30 |
| 10 | 3.35 | 1.00 | 1.67 | -0.82 | 0.90 | 0.512186 ±0.000015 | -8.82 | ±0.30 |
| 19.5 | 4.26 | 1.06 | 1.47 | -0.55 | 1.08 | 0.512191 ±0.000015 | -8.72 | ±0.30 |
| 29.5 | 5.18 | 1.05 | 1.78 | -0.55 | 1.02 | 0.512167 ±0.000015 | -9.19 | ±0.30 |
| 42.5 | 6.36 | 0.90 | 1.81 | -0.91 | 0.84 | 0.512179 ±0.000015 | -8.95 | ±0.30 |
| 52.5 | 7.28 | 0.97 | 1.55 | -0.80 | 0.80 | 0.512194 ±0.000015 | -8.66 | ±0.30 |
| 61.5 | 8.10 | 0.67 | 1.58 | -0.95 | 1.01 | 0.512197 ±0.000015 | -8.61 | ±0.30 |
| 67.5 | 9.11 | 1.13 | 1.65 | -1.07 | 0.80 | 0.512169 ±0.000015 | -9.15 | ±0.30 |
| 72.5 | 10.10 | 0.78 | 2.65 | -1.27 | 0.91 | 0.512184 ±0.000015 | -8.86 | ±0.30 |
| 77.5 | 10.33 | 0.85 | 2.75 | -1.10 | 0.83 | - | - | - |
| 81.5 | 10.37 | 0.96 | 2.41 | -1.21 | 0.96 | 0.512158 ±0.000015 | -9.36 | ±0.30 |
| 87.5 | 10.92 | 0.70 | 2.71 | -0.11 | 1.43 | 0.512184 ±0.000015 | -8.85 | ±0.30 |
| 92.5 | 11.39 | 0.77 | 3.15 | -1.00 | 0.89 | 0.512151 ±0.000015 | -9.50 | ±0.30 |
| 95.5 | 11.59 | 0.90 | 2.69 | -1.14 | 1.21 | 0.512178 ±0.000015 | -8.98 | ±0.30 |
| 100.5 | 11.92 | 0.67 | 2.78 | -0.44 | 2.11 | 0.512171 ±0.000015 | -9.11 | ±0.30 |
| 110.5 | 12.58 | 0.96 | 2.78 | -0.86 | 2.29 | 0.512189 ±0.000015 | -8.76 | ±0.30 |
| 115.5 | 12.91 | 0.71 | 2.96 | -0.54 | 2.01 | 0.512180 ±0.000015 | -8.94 | ±0.30 |
| 119.5 | 13.17 | 0.96 | 3.09 | - | - | 0.512161 ±0.000015 | -9.31 | ±0.30 |
| 129.5 | 13.83 | 0.81 | 2.99 | - | - | 0.512156 ±0.000015 | -9.41 | ±0.30 |
| 135.5 | 14.23 | 1.05 | 3.25 | -1.16 | 1.98 | 0.512167 ±0.000015 | -9.19 | ±0.30 |
| 135.5 | 14.23 | - | - | -0.94 | 2.29 | - | - | - |
| 139.5 | 14.49 | - | - | -0.96 | 2.51 | 0.512170 ±0.000015 | -9.12 | ±0.30 |
| 159.5 | 15.88 | 0.91 | 3.45 | -0.81 | 2.45 | 0.512153 ±0.000015 | -9.47 | ±0.30 |
| 169.5 | 16.54 | 1.39 | 4.09 | -0.76 | 3.47 | - | - | - |
| 179.5 | 17.20 | 1.53 | 4.30 | -0.98 | 3.45 | 0.512170 ±0.000015 | -9.12 | ±0.30 |
| 190 | 17.86 | 1.46 | 4.24 | -1.10 | 3.70 | 0.512168 ±0.000015 | -9.16 | ±0.30 |
| 198 | 18.39 | 1.65 | 4.39 | -1.24 | 3.24 | 0.512153 ±0.000015 | -9.46 | ±0.30 |
| 206 | 18.78 | 1.53 | 4.28 | -0.90 | 3.53 | - | - | - |

**Table 4**