# Peer review of "Hydrological variations of the intermediate water masses of 1 the western Mediterranean Sea during the past 20 ka inferred 2 from neodymium isotopic composition in foraminifera and 3 cold-water corals 4"

_Climate of the Past, 2016_

## Referee Comment (RC1) · Anonymous Referee #1 · 3 Aug 2016

This paper presents new an interesting data to reconstruct hydrological variations in the western Mediterranean since the Last Glacial Maximum. The study is mainly based on neodymium isotopic composition of cold-water corals from the Alboran Sea and the south Sardinian continental margin and on neodymium and stable isotopic composition of foraminifera from a sediment core recovered in the Balearic basin. This is a relevant contribution that strengthens the need of further investigations of the Mediterranean circulation, in particular during the time intervals corresponding to the sapropel S1

and ORL1 deposition, when major changes occurred. The paper is well written and structured and the study deserves publication. However, some comments may be considered in the revised version:

- From the first part of the abstract, it seems that the neodymium isotopic composition of both mixed planktonic foraminifera and cold-water corals (CWC) have been investigated at the three selected locations. This could be made clear since CWC have been analyzed only at the Alboran Sea and the south Sardinian continental margin and foraminifera only at the Balearic basin. This aspect could be clarified, also explaining how the data have been integrated. The abstract includes the main implications of the study for hydrological variations during the deposition of the S1sapropel but the data are also relevant to the deposition of the ORL1.

- The introduction could better highlight the aim of the work. Moreover, the classical references on Mediterranean climate variability are cited but more recent ones could also be included, for instance Martrat et al (2014) provide interesting high-resolution data on surface water variability of the Mediterranean Sea during the last two deglaciations, including the Holocene.

- In the material and methods section, though references are provided to get detailed information about CWC cores, additional information on core description could also be included in this paper to facilitate the whole picture of the analyzed materials. Similarly, a new core recovered in the Balearic Sea has been investigated but little is said about the description of the materials sampled except for barren of any CWC fragments. It is also mentioned that samples from this core have been used for multiproxy analyses but other than dating and estimation of SST by modern analogue techniques only neodymium and stable isotopes been analyzed so this could be better specified in section 3.1.

- Regarding the results section, there are three different subsections on core SU92-33 that may be omitted and the results could be synthetized in just one as for CWC.

Some general sentences referred to sedimentation rate as "the lowest values observed during the Holocene" could be more specific. In this section the information concerning the core MD90-917 is insufficient, it is cited as a well dated record but it is not clear if the references cited in the paragraph (line 294) are those providing the data included in Fig. 2a (in which a reference is not cited).

- The discussion is relevant and highlights the most important aspects of the hydrological variations in the Mediterranean. However, some aspects could be further discussed as the role of the eolian input in the $\varepsilon$Nd variability and why it is not affected by changes in such input. Concerning this, some additional papers on eolian input could be considered, for instance Scheuvens et al. (2013) on bulk composition of northern African dust or Rodrigo-Gamiz et al (2015) on terrigenous input provenance in the western Mediterranean. Also regarding the Nile discharge, some other recent papers could be considered as Hennekam et al (2014). In general, the results on SST are not sufficiently compared with other SST records, see for instance the previously mentioned paper from Martrat et al (2014) and also some recent papers on sea surface temperature variations in the western Mediterranean sea over the last 20 kyr (Rodrigo-Gamiz et al., 2014). It is also concluded that $\delta$18O and $\delta$13C values indicate a stratification of the water masses after 16 cal ka BP, but why the data are supporting this conclusion could be further explained in the conclusions section. The implications of the obtained results for the deposition of the ORL1 could also be included in this section.

References

- Hennekam, R., Jilbert, T., Schnetger, B., & Lange, G. J. (2014). Solar forcing of Nile discharge and sapropel S1 formation in the early to middle Holocene eastern Mediterranean. Paleoceanography, 29(5), 343-356. - Martrat, B., Jimenez-Amat, P., Zahn, R., & Grimalt, J. O. (2014). Similarities and dissimilarities between the last two deglaciations and interglaciations in the North Atlantic region. Quaternary Science Reviews, 99, 122-134. - Scheuvens, D., Schütz, L., Kandler, K., Ebert, M., & Weinbruch, S. (2013). Bulk composition of northern African dust and its source sediments—a compilation. Earth-Science Reviews, 116, 170-194. - Rodrigo-Gámiz, M., Martínez-Ruiz, F., Rampen, S. W., Schouten, S., & Sinninghe Damsté, J. S. (2014). Sea surface temperature variations in the western Mediterranean Sea over the last 20 kyr: A dual-organic proxy (UK'37 and LDI) approach. Paleoceanography, 29(2), 87-98. - Rodrigo-Gámiz, M., Martínez-Ruiz, F., Chiaradia, M., Jiménez-Espejo, F. J., & Ariztegui, D. (2015). Radiogenic isotopes for deciphering terrigenous input provenance in the western Mediterranean. Chemical Geology, 410, 237-250.

---

## Referee Comment (RC2) · Anonymous Referee #2 · 5 Aug 2016

Dubois-Dauphin et al. present new and high quality neodymium isotopic data from planktic foraminifera and scleractinian cold-water corals from three locations in the western Mediterranean. They use these data to characterise the hydrographic (not hydrological!) variability at intermediate depths in the western Mediterranean and to constrain the variability of the intermediate circulation (a key component of the Mediterranean thermohaline circulation!) in the basin through the last 20 kyr. This timespan is a valuable one, in that over the last 20 kyr the Northern Hemisphere underwent a

series of abrupt climate swings superimposed upon the transition (glacial termination I) from the Last Glacial Maximum (LGM) to the Holocene interglacial. In addition, the Mediterranean Sea featured the deposition of sapropel S1 (roughly from 10 to 6.5 ka BP) in the eastern basin and the Organic Rich Layer 1 (roughly from 14.5 to 8.5 ka BP) in the western basin and these changes appear to reflect variations in the Mediterranean circulation/ventilation (to which intermediate water circulation is central). Hence these climatic and oceanographic developments provide a rich source of information on the pattern(s) and drivers of thermohaline circulation changes in the basin. So all the ingredients are there to make the study by Dubois-Dauphin et al. a key contribution to the palaeoceanography of the Mediterranean Sea.

In summary, I think that the manuscript is certainly suited for publication in Climate of the Past, while below I identify those aspects that should be revised in order to better highlight the relevant (and novel) aspects of the study, make the data analysis/interpretation sound, strengthen the conclusions, and, in turn, make the manuscript acceptable for publication.

Major Points

Introduction. The Introduction could and should be improved and sharpened up (and the same may apply to the discussion). For example (Lines 57-65), the authors seem to build their rationale on the (potential) influence of the Mediterranean thermohaline circulation on the AMOC. But this is not the only reason for better characterising the patterns or variability and the drivers of the thermohaline circulation in this basin. The authors could also (or first) more clearly illustrate the importance of the Mediterranean circulation (an notably of the Levantine Intermediate Waters) for the deep-sea ventilation during the formation of organic-rich deposits (sapropels) across the basin (e.g., De Lange et al., 2008 – Nature Geoscience; Rohling et al., 2015 – Earth-Science Reviews and many others) and/or the more recent evidence of a link between Mediterranean circulation changes and positive phases of the North Atlantic Oscillation (e.g., Incarbona et al., 2016 – Scientific Reports). This would make the introduction section better

suited for Climate of the Past by making a more convincing case for the wide relevance of studies like the one by Dubois-Dauphin et al. to the palaeoceanography of the Mediterranean Sea and more generally to our community.

Sea Surface Temperature record. The uncertainties associated with the sea surface temperature (SST) reconstructions presented in the paper (Lines 247-255) should be quantitatively assessed. The authors state '...Reliability of SST reconstructions is estimated using a square chord distance test (dissimilarity coefficient), which represents the mean degree of similarity between the sample and the best 10 modern analogues. When the dissimilarity coefficient is lower than 0.25, the reconstruction is considered to be of good quality...'. This is a merely qualitative statement; the associated with the SST record presented in the manuscript should instead be quantified.

Data analysis. I think data generated by Dubois-Dauphin et al. are of high quality, but I also think that their analysis and presentation could and should be improved. For example, could the records in Figure 3b be stacked? This would highlight the main trends in the data and help the reader to easily follow the interpretation presented by the authors (at the moment also because of a 'wordy' and fairly unfocused discussion this is not the case). Even better, a Monte Carlo analysis of the data in which both uncertainties in the neodymium isotopes and in the chronology are considered would considerably strengthen the data analysis, allow more quantitative arguments, and make this a key example fo the use of neodymium isotopes to address palaeocirculation problems.

Data interpretation. I wonder if the data presented can be so unequivocally interpreted as a reduction of Levantine Intermediate Water (formation? circulation?) during the deposition of sapropel S1 to the extent of arguing for a circulation reversal (which most quantitative analyses so far suggest to be highly unlikely). A possibility that the data cannot rule out is that the Levantine Intermediate Water shoaled rather than weakened and the core sites were bathed by a water mass with a different isotopic fingerprint (e.g., the western Mediterranean intermediate waters proposed by the authors) because of this shoaling.

Minor Points

Lines 36-39: text is not very clear; I would recommend rewriting this bit.

Lines 272-283: I think this section can be moved to the methods and merged with sections 3.3.

Lines 483-484: What do the authors mean by 'intensity changes'?

---

## Author Comment (AC1) · 28 Sep 2016

Thank you very much for agreeing to review this paper and for your comments that have permitted to improve the quality of the manuscript.

Most of your comments have been taken into account in the revised version of the manuscript and all the proposed references have been added.

Please find below a point-by-point reply relative to your comments :

[Figure]

"From the first part of the abstract, it seems that the neodymium isotopic composition of both mixed planktonic foraminifera and cold-water corals (CWC) have been investigated at the three selected locations. This could be made clear since CWC have been analyzed only at the Alboran Sea and the south Sardinian continental margin and foraminifera only at the Balearic basin. This aspect could be clarified, also explaining how the data have been integrated. The abstract includes the main implications of the study for hydrological variations during the deposition of the S1 sapropel but the data are also relevant to the deposition of the ORL1."

-> The abstract has been revised in order to make clear the different archives and locations of the study. We have also added a sentence to explain how the data have been integrated. However, the deposition of the ORL1 being not the aim of this paper, we do not conclude on potential implications about it.

"The introduction could better highlight the aim of the work. Moreover, the classical references on Mediterranean climate variability are cited but more recent ones could also be included, for instance Martrat et al (2014) provide interesting high-resolution data on surface water variability of the Mediterranean Sea during the last two deglaciations, including the Holocene."

-> The introduction has been sharpened up following your recommendations and those of the reviewer #2. Martrat et al. (2014) has not been added in the introduction as we do not think that discussing SST variability in the introduction is relevant in our paper. However, this reference has been cited in other parts of the text.

"In the material and methods section, though references are provided to get detailed information about CWC cores, additional information on core description could also be included in this paper to facilitate the whole picture of the analyzed materials. Similarly, a new core recovered in the Balearic Sea has been investigated but little is said about the description of the materials sampled except for barren of any CWC fragments. It is also mentioned that samples from this core have been used for multiproxy

analyses but other than dating and estimation of SST by modern analogue techniques only neodymium and stable isotopes been analyzed so this could be better specified in section 3.1. "

-> Additional information on the CWC and SU92.33 cores have now been included in the text. The term "multiproxy analyses" has been replaced by "$\delta$18O, $\delta$13C and $\varepsilon$Nd analyzes".

"Regarding the results section, there are three different subsections on core SU92- 33 that may be omitted and the results could be synthetized in just one as for CWC. "

-> The subsections have been deleted and the results have been synthetized in two sections: CWC and core SU-92.33

"Some general sentences referred to sedimentation rate as "the lowest values observed during the Holocene" could be more specific. "

-> The sentence has been modified in order to better quantify the sedimentation rate.

"In this section the information concerning the core MD90-917 is insufficient, it is cited as a well dated record but it is not clear if the references cited in the paragraph (line 294) are those providing the data included in Fig. 2a (in which a reference is not cited)."

-> The reference for this core (Siani et al., 2004) has been integrated in the paragraph.

"The discussion is relevant and highlights the most important aspects of the hydrological variations in the Mediterranean. However, some aspects could be further discussed as the role of the eolian input in the $\varepsilon$Nd variability and why it is not affected by changes in such input. Concerning this, some additional papers on eolian input could be considered, for instance Scheuvens et al. (2013) on bulk composition of northern African dust or Rodrigo-Gamiz et al (2015) on terrigenous input provenance in the western Mediterranean. "

-> The role of the eolian input had been partially discussed in the text as we mentioned

the papers by Arsouze et al. (2009) and Bout-Roumazeilles et al. (2013). However, following your recommendations, we have added an additional part of the discussion based on the paper by Rodrigo-Gámiz et al. (2015) on the terrigenous input provenance in the western Mediterranean.

-> In the submitted version of the manuscript, we had cited individual references (Grousset et al., 1992, 1998; Grousset and Biscaye, 2005) that are included in the synthesis paper by Scheuvens et al. (2013). In the revised version, we have decided to remove those references and only cite "see synthesis in Scheuvens et al., 2013" to make it clear.

"Also regarding the Nile discharge, some other recent papers could be considered as Hennekam et al (2014). "

-> This reference has been added to the text of the revised version.

"In general, the results on SST are not sufficiently compared with other SST records, see for instance the previously mentioned paper from Martrat et al (2014) and also some recent papers on sea surface temperature variations in the western Mediterranean sea over the last 20 kyr (Rodrigo-Gamiz et al., 2014). "

-> The SU92-33 SST record is now compared to SST reconstructions reported in Martrat et al. (2014) and Rodrigo-Gámiz et al. (2014).

"It is also concluded that 18O and 13C values indicate a stratification of the water masses after 16 cal ka BP, but why the data are supporting this conclusion could be further explained in the conclusions section. The implications of the obtained results for the deposition of the ORL1 could also be included in this section."

-> The deposition of the ORL1 being not the aim of this paper, we do not conclude on potential implications about it.

Please also note the supplement to this comment:

http://www.clim-past-discuss.net/cp-2016-64/cp-2016-64-AC1-supplement.pdf

---

## Author Comment (AC2) · 28 Sep 2016

Thank you very much for agreeing to review this paper and for your comments that have improved the quality of the manuscript.

Most of your comments have been taken into account in the revised version of the manuscript.

Please find below a point-by-point reply relative to your comments.

[Figure]

"Introduction. The Introduction could and should be improved and sharpened up (and the same may apply to the discussion). For example (Lines 57-65), the authors seem to build their rationale on the (potential) influence of the Mediterranean thermohaline circulation on the AMOC. But this is not the only reason for better characterising the patterns or variability and the drivers of the thermohaline circulation in this basin. The authors could also (or first) more clearly illustrate the importance of the Mediterranean circulation (an notably of the Levantine Intermediate Waters) for the deep-sea ventilation during the formation of organic-rich deposits (sapropels) across the basin (e.g., De Lange et al., 2008 – Nature Geoscience; Rohling et al., 2015 – Earth-Science Reviews and many others) and/or the more recent evidence of a link between Mediterranean circulation changes and positive phases of the North Atlantic Oscillation (e.g., Incarbona et al., 2016 – Scientific Reports). This would make the introduction section better suited for Climate of the Past by making a more convincing case for the wide relevance of studies like the one by Dubois-Dauphin et al. to the palaeoceanography of the Mediterranean Sea and more generally to our community."

-> The introduction has been modified by integrating the importance of intermediate and deep water circulation during the formations of organic rich deposits. However, the evidence of a link between Mediterranean circulation changes and positive phases of the North Atlantic Oscillation has not been added as it is relevant only on a decadal timescale, which is not the target of our paper.

"Sea Surface Temperature record. The uncertainties associated with the sea surface temperature (SST) reconstructions presented in the paper (Lines 247-255) should be quantitatively assessed. The authors state ': : :Reliability of SST reconstructions is estimated using a square chord distance test (dissimilarity coefficient), which represents the mean degree of similarity between the sample and the best 10 modern analogues. When the dissimilarity coefficient is lower than 0.25, the reconstruction is considered to be of good quality: : :". This is a merely qualitative statement; the associated with the SST record presented in the manuscript should instead be quantified."

-> The uncertainties associated with SST reconstruction have been plotted on figures 2 and 3. Additional information has also been added in the Material and methods section in order to better quantify the SST reconstruction.

"Data analysis. I think data generated by Dubois-Dauphin et al. are of high quality, but I also think that their analysis and presentation could and should be improved. For example, could the records in Figure 3b be stacked? This would highlight the main trends in the data and help the reader to easily follow the interpretation presented by the authors (at the moment also because of a 'wordy' and fairly unfocused discussion this is not the case). Even better, a Monte Carlo analysis of the data in which both uncertainties in the neodymium isotopes and in the chronology are considered would considerably strengthen the data analysis, allow more quantitative arguments, and make this a key example fo the use of neodymium isotopes to address palaeocirculation problems."

-> Although both sites in the Balearic and Alboran Sea are likely bathed by the same water mass (LIW), $\varepsilon$Nd records are based on different archives (i.e. cold-water corals and planktonic foraminifera). Furthermore, the age model is different as core SU92-33 is based on 14C measurements while CWC are dated by the U-Th method. On the other hand, data obtained from CWC from the Sardinia Channel display only specific time slices instead of a continuous record over time. For these reasons, we do not think that a Monte Carlo analysis and/or a stacked record would be relevant for this study.

"Data interpretation. I wonder if the data presented can be so unequivocally interpreted as a reduction of Levantine Intermediate Water (formation? circulation?) during the deposition of sapropel S1 to the extent of arguing for a circulation reversal (which most quantitative analyses so far suggest to be highly unlikely). A possibility that the data cannot rule out is that the Levantine Intermediate Water shoaled rather than weakened and the core sites were bathed by a water mass with a different isotopic fingerprint (e.g., the western Mediterranean intermediate waters proposed by the authors) because of this shoaling."

[Figure]

-> This alternative hypothesis is now presented at the end of the discussion.

Minor Points "Lines 36-39: text is not very clear; I would recommend rewriting this bit."

-> The sentence has been slightly rephrased.

"Lines 272-283: I think this section can be moved to the methods and merged with sections 3.3."

-> This section has been re-organised following also recommendations of the reviewer #1

"Lines 483-484: What do the authors mean by 'intensity changes'?"

-> We mean changes in LIW production (enhanced or reduced). The sentence has been slightly modified to make it clear.

Please also note the supplement to this comment:
http://www.clim-past-discuss.net/cp-2016-64/cp-2016-64-AC2-supplement.pdf

———————————————————

---

## Editor Decision (ED2)

**Reply to reviewer #1 comments:**

Thank you very much for agreeing to review this paper and for your comments that have permitted to improve the quality of the manuscript.

Most of your comments have been taken into account in the revised version of the manuscript and all the proposed references have been added.

Please find below a point-by-point reply relative to your comments.

*From the first part of the abstract, it seems that the neodymium isotopic composition of both mixed planktonic foraminifera and cold-water corals (CWC) have been investigated at the three selected locations. This could be made clear since CWC have been analyzed only at the Alboran Sea and the south Sardinian continental margin and foraminifera only at the Balearic basin. This aspect could be clarified, also explaining how the data have been integrated. The abstract includes the main implications of the study for hydrological variations during the deposition of the S1 sapropel but the data are also relevant to the deposition of the ORL1.*

The abstract has been revised in order to make clear the different archives and locations of the study. We have also added a sentence to explain how the data have been integrated. However, the deposition of the ORL1 being not the aim of this paper, we do not conclude on potential implications about it.

*The introduction could better highlight the aim of the work. Moreover, the classical references on Mediterranean climate variability are cited but more recent ones could also be included, for instance Martrat et al (2014) provide interesting high-resolution data on surface water variability of the Mediterranean Sea during the last two deglaciations, including the Holocene.*

The introduction has been sharpened up following your recommendations and those of the reviewer #2. Martrat et al. (2014) has not been added in the introduction as we do not think that discussing SST variability in the introduction is relevant in our paper. However, this reference has been cited in other parts of the text.

*In the material and methods section, though references are provided to get detailed information about CWC cores, additional information on core description could also be included in this paper to facilitate the whole picture of the analyzed materials. Similarly, a new core recovered in the Balearic Sea has been investigated but little is said about the description of the materials sampled except for barren of any CWC fragments. It is also mentioned that samples from this core have been used for multiproxy analyses but other than dating and estimation of SST by modern analogue techniques only neodymium and stable isotopes been analyzed so this could be better specified in section 3.1.*

Additional information on the CWC and SU92.33 cores have now been included in the text. The term "multiproxy analyses" has been replaced by "$\delta^{18}O$, $\delta^{13}C$ and $\epsilon Nd$ analyzes".

*Regarding the results section, there are three different subsections on core SU92-33 that may be omitted and the results could be synthetized in just one as for CWC.*

The subsections have been deleted and the results have been synthetized in two sections: CWC and core SU-92.33

*Some general sentences referred to sedimentation rate as "the lowest values observed during the Holocene" could be more specific.*

The sentence has been modified in order to better quantify the sedimentation rate.

*In this section the information concerning the core MD90-917 is insufficient, it is cited as a well dated record but it is not clear if the references cited in the paragraph (line 294) are those providing the data included in Fig. 2a (in which a reference is not cited).*

The reference for this core (Siani et al., 2004) has been integrated in the paragraph.

*The discussion is relevant and highlights the most important aspects of the hydrological variations in the Mediterranean. However, some aspects could be further discussed as the role of the eolian input in the εNd variability and why it is not affected by changes in such input. Concerning this, some additional papers on eolian input could be considered, for instance Scheuvens et al. (2013) on bulk composition of northern African dust or Rodrigo-Gamiz et al (2015) on terrigenous input provenance in the western Mediterranean.*

The role of the eolian input had been partially discussed in the text as we mentioned the papers by Arsouze et al. (2009) and Bout-Roumazeilles et al. (2013). However, following your recommendations, we have added an additional part of the discussion based on the paper by Rodrigo-Gámiz et al. (2015) on the terrigenous input provenance in the western Mediterranean.

In the submitted version of the manuscript, we had cited individual references (Grousset et al., 1992, 1998; Grousset and Biscaye, 2005) that are included in the synthesis paper by Scheuvens et al. (2013). In the revised version, we have decided to remove those references and only cite "see synthesis in Scheuvens et al., 2013" to make it clear.

*Also regarding the Nile discharge, some other recent papers could be considered as Hennekam et al (2014).*

This reference has been added to the text of the revised version.

*In general, the results on SST are not sufficiently compared with other SST records, see for instance the previously mentioned paper from Martrat et al (2014) and also some recent papers on sea surface temperature variations in the western Mediterranean sea over the last 20 kyr (Rodrigo-Gamiz et al., 2014).*

The SU92-33 SST record is now compared to SST reconstructions reported in Martrat et al. (2014) and Rodrigo-Gámiz et al. (2014).

*It is also concluded that 18O and 13C values indicate a stratification of the water masses after 16 cal ka BP, but why the data are supporting this conclusion could be further explained in the conclusions section. The implications of the obtained results for the deposition of the ORL1 could also be included in this section.*

The deposition of the ORL1 being not the aim of this paper, we do not conclude on potential implications about it.

**Reply to reviewer #2 comments:**

Thank you very much for agreeing to review this paper and for your comments that have improved the quality of the manuscript.

Most of your comments have been taken into account in the revised version of the manuscript.

Please find below a point-by-point reply relative to your comments.

*Introduction. The Introduction could and should be improved and sharpened up (and the same may apply to the discussion). For example (Lines 57-65), the authors seem to build their rationale on the (potential) influence of the Mediterranean thermohaline circulation on the AMOC. But this is not the only reason for better characterising the patterns or variability and the drivers of the thermohaline circulation in this basin. The authors could also (or first) more clearly illustrate the importance of the Mediterranean circulation (an notably of the Levantine Intermediate Waters) for the deep-sea ventilation during the formation of organic-rich deposits (sapropels) across the basin (e.g., De Lange et al., 2008 – Nature Geoscience; Rohling et al., 2015 – Earth-Science Reviews and many others) and/or the more recent evidence of a link between Mediterranean circulation changes and positive phases of the North Atlantic Oscillation (e.g., Incarbona et al., 2016 – Scientific Reports). This would make the introduction section better suited for Climate of the Past by making a more convincing case for the wide relevance of studies like the one by Dubois-Dauphin et al. to the palaeoceanography of the Mediterranean Sea and more generally to our community.*

The introduction has been modified by integrating the importance of intermediate and deep water circulation during the formations of organic rich deposits. However, the evidence of a link between Mediterranean circulation changes and positive phases of the North Atlantic Oscillation has not been added as it is relevant only on a decadal timescale, which is not the target of our paper.

*Sea Surface Temperature record. The uncertainties associated with the sea surface temperature (SST) reconstructions presented in the paper (Lines 247-255) should be quantitatively assessed. The authors state ': : :Reliability of SST reconstructions is estimated using a square chord distance test (dissimilarity coefficient), which represents the mean degree of similarity between the sample and the best 10 modern analogues. When the dissimilarity coefficient is lower than 0.25, the reconstruction is considered to be of good quality: : :". This is a merely qualitative statement; the associated with the SST record presented in the manuscript should instead be quantified.*

The uncertainties associated with SST reconstruction have been plotted on figures 2 and 3. Additional information has also been added in the Material and methods section in order to better quantify the SST reconstruction.

*Data analysis. I think data generated by Dubois-Dauphin et al. are of high quality, but I also think that their analysis and presentation could and should be improved. For example, could the records in Figure 3b be stacked? This would highlight the main trends in the data and help the reader to easily follow the interpretation presented by the authors (at the moment also because of a 'wordy' and fairly unfocused discussion this is not the case). Even better, a Monte Carlo analysis of the data in which both uncertainties in the neodymium isotopes and in the chronology are considered would considerably strengthen the data analysis, allow*

*more quantitative arguments, and make this a key example fo the use of neodymium isotopes to address palaeocirculation problems.*

Although both sites in the Balearic and Alboran Sea are likely bathed by the same water mass (LIW), εNd records are based on different archives (i.e. cold-water corals and planktonic foraminifera). Furthermore, the age model is different as core SU92-33 is based on $^{14}$C measurements while CWC are dated by the U-Th method.
On the other hand, data obtained from CWC from the Sardinia Channel display only specific time slices instead of a continuous record over time.
For these reasons, we do not think that a Monte Carlo analysis and/or a stacked record would be relevant for this study.

*Data interpretation. I wonder if the data presented can be so unequivocally interpreted as a reduction of Levantine Intermediate Water (formation? circulation?) during the deposition of sapropel S1 to the extent of arguing for a circulation reversal (which most quantitative analyses so far suggest to be highly unlikely). A possibility that the data cannot rule out is that the Levantine Intermediate Water shoaled rather than weakened and the core sites were bathed by a water mass with a different isotopic fingerprint (e.g., the western Mediterranean intermediate waters proposed by the authors) because of this shoaling.*

This alternative hypothesis is now presented at the end of the discussion.

*Minor Points*
*Lines 36-39: text is not very clear; I would recommend rewriting this bit.*

The sentence has been slightly rephrased.

*Lines 272-283: I think this section can be moved to the methods and merged with sections 3.3.*

This section has been re-organised following also recommendations of the reviewer #1

*Lines 483-484: What do the authors mean by 'intensity changes'?*

We mean changes in LIW production (enhanced or reduced). The sentence has been slightly modified to make it clear.

[revised manuscript text omitted]

---

## Author Response (AR3)

Dear Editor,

Thank you very much for reading this paper. We took account of your minor revision in the last version of the manuscript.

The text has been modified following your recommendations. Only for the "radiocarbon dating" section we did not add information about the interpolation as we write later in the "result" section : "*The age model for the upper 1.2 m of the core SU92-33 was based on 7 AMS-14C age measurements and a linear interpolation between these ages (Table 3, Fig. 2). For the lower portion of the core, a control point was established at the onset of the last deglaciation, which is coeval in the western and central Mediterranean Sea at ~17 cal ka BP (Sierro et al., 2005; Melki et al., 2009; Siani et al., 2001)". (lines 303-306).*

All the references in preparation have been replaced by personal communication.

Figure 3 has been modified following your recommendations. The y-axis for the εNd record from the Balearic-Alboran Seas has been resized. Plot a) is now renamed "Balearic Sea" and plot b is now renamed "Alboran and Balearic Seas"

[revised manuscript text omitted]